# Modification of High-Density Lipoprotein Functions by Diet and Other Lifestyle Changes: A Systematic Review of Randomized Controlled Trials

**DOI:** 10.3390/jcm10245897

**Published:** 2021-12-15

**Authors:** Albert Sanllorente, Camille Lassale, Maria Trinidad Soria-Florido, Olga Castañer, Montserrat Fitó, Álvaro Hernáez

**Affiliations:** 1Cardiovascular Risk and Nutrition Research Group, Hospital del Mar Medical Research Institute (IMIM), 08003 Barcelona, Spain; albertsanllorente@gmail.com (A.S.); ocastaner@imim.es (O.C.); 2PhD Program in Biomedicine, Universitat Pompeu Fabra, 08003 Barcelona, Spain; 3Consorcio CIBER, M.P. Fisiopatología de la Obesidad y Nutrición (CIBEROBN), Instituto de Salud Carlos III, 28029 Madrid, Spain; classale@imim.es; 4Cardiovascular Epidemiology and Genetics, Hospital del Mar Medical Research Institute (IMIM), 08003 Barcelona, Spain; 5Biomedical Nutrition, Pure and Applied Biochemistry, Lund University, 222 00 Lund, Sweden; mariat.soria@gmail.com; 6Centre for Fertility and Health, Norwegian Institute of Public Health, 0473 Oslo, Norway; 7Blanquerna School of Health Sciences, Universitat Ramon Llull, 08025 Barcelona, Spain

**Keywords:** high-density lipoprotein, fatty acids, antioxidants, ethanol, physical activity, trials

## Abstract

High-density lipoprotein (HDL) functional traits have emerged as relevant elements that may explain HDL antiatherogenic capacity better than HDL cholesterol levels. These properties have been improved in several lifestyle intervention trials. The aim of this systematic review is to summarize the results of such trials of the most commonly used dietary modifications (fatty acids, cholesterol, antioxidants, alcohol, and calorie restriction) and physical activity. Articles were screened from the Medline database until March 2021, and 118 randomized controlled trials were selected. Results from HDL functions and associated functional components were extracted, including cholesterol efflux capacity, cholesteryl ester transfer protein, lecithin-cholesterol acyltransferase, HDL antioxidant capacity, HDL oxidation status, paraoxonase-1 activity, HDL anti-inflammatory and endothelial protection capacity, HDL-associated phospholipase A2, HDL-associated serum amyloid A, and HDL-alpha-1-antitrypsin. In mainly short-term clinical trials, the consumption of monounsaturated and polyunsaturated fatty acids (particularly omega-3 in fish), and dietary antioxidants showed benefits to HDL functionality, especially in subjects with cardiovascular risk factors. In this regard, antioxidant-rich dietary patterns were able to improve HDL function in both healthy individuals and subjects at high cardiovascular risk. In addition, in randomized trial assays performed mainly in healthy individuals, reverse cholesterol transport with ethanol in moderate quantities enhanced HDL function. Nevertheless, the evidence summarized was of unclear quality and short-term nature and presented heterogeneity in lifestyle modifications, trial designs, and biochemical techniques for the assessment of HDL functions. Such findings should therefore be interpreted with caution. Large-scale, long-term, randomized, controlled trials in different populations and individuals with diverse pathologies are warranted.

## 1. Introduction

Low concentrations of high-density lipoprotein (HDL) cholesterol (HDL-C) have been linked with greater incidence of coronary heart disease in epidemiological studies [1,2]. Nevertheless, experimental and genetic studies have questioned the therapeutic utility of raising HDL-C levels. Pharmacological interventions aimed at increasing HDL-C concentrations (fibrates, niacin, statins, inhibitors of the cholesteryl ester transfer protein –CETP–) have failed to reduce the incidence of cardiovascular disease (CVD) [1,3]. In addition, Mendelian randomization studies have reported that presenting genetic predisposition to high levels of HDL-C is not linked to lower CVD risk [4,5], although recent results show a likely causal association with medium-size HDL-C [6].

On the other hand, HDL functions and associated functional components have been shown to be independently associated with lower CVD incidence [7] and stand as promising alternative biomarkers to explain the HDL atheroprotective role. The most studied HDL atheroprotective function is reverse cholesterol transport (Figure 1), which can be measured in vitro by the cholesterol efflux capacity (CEC) technique. It evaluates the ability of HDLs to remove cholesterol excess from cells and is measured in macrophage-derived cell cultures incubated with radio-labelled or fluorescent-labelled cholesterol [8]. HDLs can also be linked with enzymes related to HDL reverse cholesterol transport, such as lecithin cholesterol acyltransferase (LCAT), involved in cholesterol esterification, or cholesteryl ester transfer protein (CETP), crucial for cholesterol removal to the liver (see Figure 1) [8]. The second HDL atheroprotective function is antioxidant capacity, the ability to prevent LDL oxidation [9]. HDL carries antioxidant enzymes capable of degrading oxidized lipids, mainly paraoxonase-1 (PON1) and phospholipase A2 (LpPLA2) [9]. The global antioxidant capacity of HDLs can be measured in vitro by techniques such as the HDL oxidative/inflammatory index (HOII) [10]. On the contrary, HDLs could become dysfunctional after oxidation of their components. The oxidative status of HDLs can be evaluated by measuring the content of malondialdehyde (a lipid peroxide) [11]. A third HDL function is the capacity to modulate inflammatory responses. HDLs are potentially able to decrease expression of endothelial adhesion proteins and chemokines [12]. However, HDLs can also carry on their surface pro-inflammatory proteins related to HDL dysfunctionality, such as serum amyloid A (SAA) and alpha-1-antitrypsin [13]. Finally, this lipoprotein could also present a protective effect on the endothelial layer of the arteries [12]. A healthy endothelium maintains a proper permeability and regulates vascular tone, helping to prevent atherosclerosis. In this regard, HDLs have shown the capacity to improve the release of nitric oxide, a potent vasodilator secreted by the endothelium. Nitric-oxide production can be evaluated in vitro in cellular cultures of endothelial cells [12].

In a recent meta-analysis, high CEC values have been inversely related to CVD incidence [7]. Furthermore, CVD incidence has been linked to HDL antioxidant/anti-inflammatory properties and HDL-related endothelial protection in individual prospective studies in a general population and one at elevated cardiovascular disease risk [14,15,16]. There is a wide range of lifestyle intervention studies regarding other HDL functions. They include interventions on several HDL functions with different dietary modifications (intake of monounsaturated, polyunsaturated, saturated, and trans fatty acids–MUFAs, PUFAs, SFAs, and TFAs, respectively–cholesterol, antioxidant vitamins, phenolic compounds and other minor compounds, ethanol, and calorie restriction) and physical activity. The aim of this systematic review is to summarize all the evidence of the effects of dietary modifications and physical activity on HDL functions.

## 2. Materials and Methods

### 2.1. Search Strategy

To identify relevant trials, we searched the electronic database PubMed, looking for articles published until 10 March 2021. We performed twelve searches, one for each HDL function or HDL-associated functional component terms; (1) CEC activity; (2) CETP activity; (3) lecithin cholesterol acyl transferase (LCAT) activity; (4) HDL antioxidant capacity; (5) HDL oxidation status; (6) PON1 activity; (7) HDL anti-inflammatory and endothelial protection properties; (8) HDL-associated phospholipase A2; (9) HDL-associated SAA; (10) HDL sphingosine-1-phosphate content; (11) HDL-alpha-1-antitrypsin; and (12) HDL-associated complement proteins. The exact search terms in each case are displayed in the Appendix A.

The literature search was developed by two authors (A.S. and A.H.), with a standardized strategy. The articles were first filtered according to title, then abstract, and finally full text content. The bibliography of each selected article was also reviewed to find additional references. Any disagreement between the two authors was resolved by a third author (M.F.).

### 2.2. Study Selection, Inclusion, and Exclusion Criteria

Studies included were randomized controlled trials (RCTs) with any lifestyle intervention in humans that modified HDL functional traits (including whole diets, individual dietary components or near-dietary dose supplements, and physical activity). Postprandial studies, interventions of less than a week, studies with fewer than ten participants, and studies that used pharmacological therapies or high-dose supplements were discarded. The search was limited to articles written in English.

### 2.3. Data Extraction

The following information was extracted from each article: author, year, country, number of participants and characteristics (basal disease), type of intervention (randomization, control group, dose, and duration), parameter measured, and study outcomes. The study outcomes were quite heterogeneous due to the diversity of methodologies employed to measure HDL functions. All methodologies are stated in the beginning of each section of the review.

To facilitate analyses, lifestyle interventions were sub-grouped into five categories: dietary lipid interventions, antioxidant-rich interventions, ethanol, physical activity and calorie-restriction interventions, and other lifestyle interventions.

### 2.4. Risk of Bias Assessment

The risk of bias of each RCT was assessed by two authors (A.S. and A.H.) with the Cochrane Collaboration Risk of Bias Tool [17]. This instrument includes seven domains: (1) random sequence generation; (2) allocation concealment; (3) blinding of participants and personnel; (4) blinding of outcome assessment; (5) incomplete outcome data; (6) selective reporting; and (7) other potential biases. We individually assessed and classified each domain as low, high, or unclear risk of bias (when the information provided was insufficiently clear). The overall risk of an article was evaluated as low when the majority of the domains corresponded to low risk and no key domain was high-risk; as high if one or more key domains were classified as high-risk; and unclear when the majority of the key domains were categorized as unclear risk of bias.

## 3. Results

### 3.1. Study Selection and Description

We obtained 12,796 results from all twelve searches combined. After screening the primary search and excluding duplicates, 815 articles were selected. Following screening by abstract, 263 were considered adequate for full-text review. Finally, after examining the full text, 88 articles were excluded for not following a randomized design, 18 for using pharmacological therapies in combination with lifestyle interventions, 24 for not studing any pre-specified HDL function technique, eight for employing vitamins at considerably high dosage, and seven for being postprandial interventions, shorter than a week, or including fewer than 10 participants. Finally, a total of 118 RCTs were selected for a qualitative synthesis (Figure 2).

Cholesterol efflux was studied in 37 of the trials, HDL CETP activity or mass in 53, LCAT activity in 32, antioxidant properties in six, HDL oxidation in six, PON1 activity in 45, HDL anti-inflammatory and endothelial protection properties in nine, HDL-associated phospholipase A2 in one, HDL-bound SAA in two, and HDL-alpha-1-antitrypsin in one article. No results were found for HDL sphingosne-1-phosphate and HDL-associated complement proteins.

### 3.2. Characteristics of Studies Included

All studies were RCTs (with a parallel group or crossover design) and included a total of 5645 participants. Most were short-term interventions, ranging from 7 days to 6 months. Only 10 RCTs (8.4% of the total) [18,19,20,21,22,23,24,25,26,27,28] lasted from 6 to 12 months. In addition, sample sizes were generally modest. With respect to number of participants, 73.4% of the studies included 50 or fewer subjects, 17.8% had between 51 and 100, and 8.4% (ten studies) include more than 100 subjects [19,20,21,22,25,28,29,30,31,32]. Due to the nature of lifestyle interventions, most of the studies were not blinded to participants.

### 3.3. Quality Assessment

From 118 RCTs, 28.0% presented a low risk of bias, 17.8% a high risk, and 54.2% an unclear risk (Figure 3). Unclear risk was mainly caused by the absence of detailed information in the study protocols. Individual bias evaluation of the articles is available in Appendix A.

### 3.4. Dietary Lipids and HDL Function

#### 3.4.1. Monounsaturated Fatty Acids (MUFA): Oleic-Acid-Rich Oils

To determine the effect of MUFA interventions on HDL functions, nine studies were selected (Table 1). All of them assessed the effect of MUFAs from olive oil, canola oil, and peanut oil, compared to saturated fats and PUFAs.

In a 4-week RCT, MUFA intake (olive oil) was associated with a 4.7% increase in CEC values compared to an equivalent fat-calorie intake from cheese [33]. Similarly, an RCT with a 4-week consumption of canola oil (59% MUFAs) or a high-oleic canola oil (72% MUFAs) increased CEC by 39.1% and 33.6%, respectively, relative to baseline levels but not when compared to three interventions rich in PUFAs [29]. In contrast, an 8-week oleic-acid-rich diet did not improve CEC relative to a linoleic-acid-rich diet [34].

In one RCT, MUFA intake was associated with decreased CETP activity relative to SFA-rich and TFA-rich diets of 6 weeks [35]. However, no significant effects were reported in another RCT comparing a 35-day diet rich in canola oil (49% MUFAs) to diets rich in palm oil (50% SFAs), soybean oil (44% PUFAs), and partially hydrogenated soybean oil (13% TFAs) [36].

A MUFA-rich 6-week RCT based on the consumption of peanut oil increased serum LCAT activity relative to diets rich in rapeseed oil (also MUFA-rich) and dairy fats (from butter and cream, SFA-rich) [37]. In contrast, LCAT activity was not modified by a MUFA-rich dietary intervention compared to two interventions rich in PUFA oils during a period of 2 weeks [38]. HDL particles did not present higher levels of LCAT on their surface after a 32-day MUFA-rich diet intervention, compared to a high carbohydrate diet [39].

Finally, in relation to oxidative-stress-related properties, a MUFA-based supplemented intervention was not linked to changes in PON1 paraoxonase activity compared to supplements of EPA and DHA (2 g/day) during a 6-week period [40]. Nevertheless, overall HDL oxidation was decreased relative to an 8-week linoleic-rich diet [34].

From the nine studies regarding MUFA intervention, five reported no effect on HDL function biomarkers at all (four of them, however, were compared with PUFA interventions), two observed some effect in healthy volunteers, and two more found some effect in obese and metabolic syndrome patients.

#### 3.4.2. Polyunsaturated Fatty Acids: Vegetable Oils and Nuts

Twenty-one RCTs evaluated the effect of vegetable oils, linoleic-rich nuts, and linolenic fatty acids on HDL functions (Table 2). Seven assessed the effects of vegetable PUFAs against SFA and TFA interventions, six against other PUFA types and MUFAs, and seven compared them to low-fat diets and PUFA-free diets.

Regarding cholesterol metabolism, the consumption of almonds (43 g/day for 6 weeks) increased non-ABCA1 CEC compared to an isocaloric muffin without PUFAs in normal-weight participants with non-significant results for global and ABCA1 CEC [41]. A 4-week intervention with corn and safflower oils (both rich in linoleic acid) and an intervention with flaxseed oil (rich in linolenic acid) incremented CEC values relative to baseline [29]. Another RCT with two servings/day of pistachios for 4 weeks was associated with CEC improvements relative to a subgroup of participants with low C-reactive protein levels consuming one serving/day [42]. However, another four RCTs assessing vegetal PUFA intake did not report associations with changes in CEC relative to MUFA-rich diets [34,43], nor SFA- or TFA-rich diets [44,45].

Regarding CETP, a flaxseed-oil intervention (α-linolenic acid, 5.5 g/day for 12 weeks) decreased CETP activity relative to the control diet with corn oil [46]. A safflower-oil-rich diet for 6 weeks (50% of total dietary fat, rich in linoleic acid) decreased CETP from HDLs to apolipoprotein B-containing lipoproteins in relation to a butter-rich diet [47]. A 17-day linoleic-acid-rich diet (8% of total energy) decreased CETP activity relative to a TFA-rich diet [48]. An 8-week baru-almond-enriched diet (30.13% of fat as linoleic acid) decreased plasma CETP levels [49]. However, other trials did not report any changes in CETP activity or concentrations after an intervention with pistachios [50], as well as two PUFA-rich diets, relative to SFA-rich dietary arms [51,52].

LCAT activity increased after a 6-week diet complemented with 60 mL/day of sunflower oil (rich in linoleic acid) compared to a low-erucic-acid rapeseed-oil diet (rich in MUFAs) and a diet rich in dairy fats (SFAs) [37]. Nevertheless, LCAT activity in plasma was decreased in relation to baseline values after 2 weeks of 60 g/day of flaxseed oil (rich in α-linolenic acid) [38], although no changes were reported in a similar trial [53].

Finally, when assessing HDL antioxidant properties, the consumption of a walnut-paste-enriched meat (20% walnut paste, 750 g/week) increased PON1 paraoxonase activity relative to the control intervention with a low-fat diet [54], as well as relative to baseline in two similar RCTs with the same walnut intake [55,56]. Intake of linoleic-acid-rich safflower oil (4.5 g/day for 4-weeks) was also associated with increases in PON arylesterase activity relative to trans-conjugated linoleic-acid intervention in an RCT [57].

Briefly, in healthy individuals, no effect of a PUFA intervention (vegetable oils and nuts) on HDL function was observed in four out of seven studies, while in three studies, some effect was reported. From 14 studies in subjects with cardiovascular risk-related pathologies, 10 demonstrated some effect, while no benefit was reported in four.

#### 3.4.3. Polyunsaturated Fatty Acids: Fish, Eicosapentaenoic and Docosahexaenoic Acids (EPA, DHA)

The effect of fish fatty acids (eicosapentaenoic and docosahexaenoic acids (EPA, DHA)) on HDL functions was evaluated in eleven RCTs (Table 3). Three interventions made comparisons with other PUFAs, and eight were supplements compared with placebos or non-supplemented interventions.

An intervention with a DHA-enriched canola-oil smoothie (5.8% DHA content) was associated with a 55% increment in CEC values, relative to baseline, in metabolic syndrome patients [29]. However, a trial studying the effect of dietary fatty fish (1 g/day EPA + DHA) compared to lean fish or with camelina oil (high in linolenic) did not show effects on CEC [58]. Additionally, higher doses of EPA + DHA in supplements (four capsules of 1.88 g EPA + 1.48 g of DHA per day) did not result in effects on CEC compared to a placebo [59].

Cholesteryl ester transference from HDLs to apolipoprotein B-containing lipoproteins decreased relative to baseline after an EPA + DHA supplement (4 g/day) [60]. In contrast, CETP levels did not change in an RCT with an EPA + DHA supplement (four capsules 1.88 g EPA + 1.48 g DHA per day) [59]. LCAT activity was also decreased relative to baseline values in an RCT after a fish-oil supplement (3.8 g/day) [53], whilst the circulating levels of the enzyme did not change in RCTs assessing an EPA + DHA supplement [59], and in lower doses in an EPA/DHA-enriched milk (131.25 mg EPA and 243.75 mg DHA) [61].

With respect to HDL antioxidant properties, PON1 activity was significantly increased relative to a placebo control in RCTs with fish-oil supplements (1 g/day) [62] and EPA (2 g/day). PON1 circulating levels also incremented in RCTs with 2 g/day EPA supplements [63], compared to the control, and with EPA/DHA-enriched milk (relative to baseline) [61]. However, no significant differences in PON1 levels were found in an RCT studying a low-dose supplement of DHA (500 mg/day) [64] and after the consumption of 2 g/day purified EPA and DHA relative to olive oil [40]. The global antioxidant capacity of HDLs was evaluated in an intervention with EPA + DHA supplementation (4 g/day), which was associated with a deleterious effect on the HDL inflammatory index in heart failure patients compared to lesser doses of EPA + DHA (1 g/day) and placebo treatments [65].

From the 11 studies of patients with cardiovascular risk-related pathologies, rheumatoid arthritis or heart failure, seven showed some effect on HDL function after a fish, EPA, or DHA intervention, whilst four reported no effects.

#### 3.4.4. Saturated (SFA) and Trans Fatty Acids (TFA)

Sixteen RCTs evaluated the effect of SFAs and TFAs on HDL functions (Table 4). Among them, twelve were compared with PUFAs and MUFAs.

Regarding CEC, a diet containing 12.5% SFAs from butter improved CEC values (+4.3%) relative to a cheese-rich isocaloric diet or a carbohydrate-rich diet [33], whilst no differences relative to baseline were observed after a 4-week intervention with palm-oil-rich and TFA-rich (hydrogenated soybean oil) diets in healthy individuals [44].

Increases in CETP activity and mass were reported in an RCT with a palmitic-acid-enriched diet (45% total fat content) [35], compared with MUFA-rich dietary patterns. Rises in CETP activity were also found in RCTs comparing a butter-rich diet with a linoleic-acid-rich intervention [47] and a lauric-acid-rich diet (relative to baseline) [66]. In contrast, no effects were observed in two short-term RCTs after consumption of palm-oil-rich diets relative to two PUFA-rich diets [36,51] and in a general saturated-fat-rich diet relative to a PUFA-rich diet [52]. Another RCT comparing the effect of 70 g/day saturated medium-chain fatty acids (caprylic and capric acids) with a high-MUFA intervention (70 g/day) did not report changes in CETP activity [67].

Intake of TFAs was associated with an increase in CETP activity when a 17-day TFA-rich dietary intervention (8% trans fats) was compared with stearic and linoleic-acid-rich diets [48]. In a similar manner, a 5-week margarine-rich diet intervention also incremented CETP activity relative to butter-rich and semiliquid margarine-rich diets [68], and a marginally significant increase was reported after 5 weeks high-TFA margarine consumption in older hypercholesterolemic women [69]. In comparison, no effects on CETP were observed after a 5-week consumption of partially hydrogenated soybean oil in hypercholesterolemic individuals [36,70] and another TFA-rich intervention [71].

Regarding LCAT activity, 6-week SFA-rich diets (from butter and cream) decreased LCAT activity relative to sunflower and peanut oils (rich in linoleic and oleic acids) [37]. However, no effects were reported after a 5-week TFA-rich dietary modification (based on partially hydrogenated soybean oil with 3% TFAs) [70].

Finally, when referring to HDL antioxidant properties, a 4-weekTFA-rich intervention (9.3% total fat in TFA from margarine) decreased PON1 paraoxonase activity relative to a SFA-rich diet [72]. However, no effects were reported after a 5-week consumption of a partially hydrogenated soybean oil (13% total fats as TFAs) or palm oil (50% total fats as SFAs) [36].

From 16 studies with SFA and TFA interventions, eight were conducted with healthy volunteers, and some detrimental effect on HDL function was reported in four of them. In addition, in subjects with dyslipidemia or obesity, some effect was observed in four studies.

#### 3.4.5. Dietary Cholesterol

The effects of dietary cholesterol, provided mainly by egg intake, on HDL functions were studied in 15 RCTs (Table 5).

Consuming two eggs/day significantly incremented CEC relative to control interventions (a low-fat diet with the same weight of yolk-free eggs) after 24 days and 4 weeks, respectively [73,74], and a similar effect relative to baseline values was described after a 12-week intervention with three eggs/day [75].

Three RCTs increased CETP activity after the consumption of one egg/day (64 mg/day) per 1 month in cholesterol-sensitive participants (individuals whose plasma total and HDL-C levels increased after egg consumption) in relation to non-egg consumption [76,77] and relative to baseline levels [78]. Increases in CETP mass (also parallel to increases in HDL-C levels) were observed in a high-cholesterol diet (320 mg/1000 kcal), relative to a low-cholesterol diet (80 mg/1000 kcal) [79] and after increasing egg consumption in men [80] but not in women [81]. Conversely, no effects were reported in four RCTs based on the consumption of three eggs/day for 12 weeks in metabolic syndrome patients [82], after two eggs/day for 4 weeks in overweight postmenopausal women [74], in heathy young participants [83], and after a one egg/day intervention for 4 weeks in postmenopausal women [84].

LCAT activity incremented parallel to increases in cholesterol intake. In one RCT, it augmented in interventions based on one and two eggs/day for a month (relative to lower egg intake) in healthy young individuals [85], three eggs/day for 12 weeks in carbohydrate-restricted diets in metabolic syndrome and excess-weight patients (relative to baseline levels) [82,86], and one egg/day for 1 month in dietary-cholesterol-sensitive individuals [77,78]. No effects were found in a 4-week two-egg intervention [74,83] and in a one-egg intervention after a month [84].

Finally, in some RCTs, the consumption of eggs did not modify HDL antioxidant capacity or PON10-circulating activity [74,87]. It was, however, associated with HDLs with greater lipid hydroperoxide content (more oxidized) and pro-inflammatory proteins (serum amyloid A) [87].

Briefly, from the 11 studies of healthy subjects, eight of them reported some effect. Four more studies were performed with subjects with some pathology (overweight/obese, metabolic syndrome), and they all observed some effect on HDL function after an egg-intake intervention.

### 3.5. Antioxidants and HDL Function

#### 3.5.1. Antioxidant Nutrients and Antioxidant-Rich Foods

The capacity of polyphenols, carotenoids, and antioxidant vitamins in foods and supplements to improve HDL functions was assessed in 26 RCTs (Table 6).

A 3-week intervention with a virgin olive oil naturally rich in phenolic compounds (25 mL/day, raw) increased CEC by 3% relative to the control intervention (a refined olive oil) [88], and a functional virgin olive oil, further enriched with thyme polyphenols (25 mL/day, consumed raw) also augmented CEC relative to a standard virgin olive oil after 3 weeks [89]. Supplements of anthocyanins (320 mg/day) for 12 and 24 weeks were associated with significant increases in CEC (by 18–20%) relative to placebo [25,31]. However, a 3-week intervention based on an anthocyanin-rich grape powder (60 g/day) did not lead to changes in metabolic syndrome patients [90].

CETP activity was also investigated in several short-term antioxidant-based RCTs. Lycopene supplementation (70 mg/week) for 12 weeks was associated with decreased CETP activity relative to a placebo [91]. A similar effect was described after 12-week anthocyanin supplementation (320 mg/day) [31] and the consumption of a lyophilized grape powder (6 g) for 4 weeks [92]. However, no effects were observed after a 6-week intake of one cup of raisins per day [93]. The use of French-press coffee (high in diterpenes, such as cafestol and kahweol) increased CETP activity compared to a control filtered coffee (less diterpene content) [94]. Only two studies, one assessing the effects of phenol-enriched functional virgin olive oils and the other, a dietary intervention with vegetables, berries, and apples, reported no changes in CETP activity [95,96].

Finally, LCAT activity also appears to improve after antioxidant-rich interventions. High intake of lycopene (such as a lycopene-rich diet 224 to 350 mg/week or supplementation of 70 mg/week for 12 weeks) was associated with increased LCAT activity relative to baseline [91]. A phenolic-compound-rich, functional virgin olive oil also augmented LCAT mass relative to the control intervention (with standard virgin olive oil). It induced the changes in HDL composition that are usually associated with LCAT function (relative decreases in free cholesterol in the lipoprotein) [95]. Two interventions with diterpenes from coffee (cafestol and kahweol) showed decreased LCAT activity relative to baseline levels [94,97]. In contrast, an anthocyanin supplement and a dietary intervention with vegetables, berries, and apples were not related to improvements in the function of this enzyme [31,96].

In controlled trials, dietary antioxidants have also been associated with beneficial effects on HDLs by incrementing content and making HDLs more oxidation-resistant. Two interventions with natural and functional virgin olive oils also increased HDL content of dietary antioxidants (olive-oil phenolic compound metabolites, β-cryptoxanthin, and lutein) relative to the control interventions [88,89]. Indeed, as many as 17 clinical trials have focused on the promotion of PON1 antioxidant activities. Regarding carotenoids, high intake of lycopene (lycopene-rich diet 224 to 350 mg/week or supplementation of 70 mg/week) increased PON1 arylesterase activity relative to the control [91], and a supplement of astaxanthin (4 mg) for 3 months increased PON1 diazoxonase activity relative to baseline [98]. However, two trials with tomato juice (lycopene-rich 37–47 mg/day) for 2 and 8 weeks did not show differences relative to control interventions (water and carrot juice) [99,100]. In relation to phenolic compounds, two anthocyanin-based interventions (anthocyanin supplementation of 320 mg/d for 24 weeks, barberry juice 200 mL/day for 8 weeks) increased PON1 paraoxonase activity and concentration, respectively [25,101], and a 6-month intervention with a pomegranate extract (1 g/day) incremented PON1 lactonase activity relative to a placebo [24]. In contrast, a 3-week intervention with grape powder (60 g/day) did not result in significant differences in PON1 activity [90]. In three 3-week RCTs, two phenolic-compound-rich oils (virgin olive and argan oils) significantly increased PON1 paraoxonase and arylesterase activities [102]. Moreover, two virgin olive oils (one natural and the other enriched with olive phenolic compounds) promoted PON1 paraoxonase- and lactonase-specific activities [103], and another virgin olive oil enriched with thyme polyphenols boosted arylesterase activity [95], all relative to baseline values. With regard to other phenolic-compound-rich interventions in RCTs, powdered ginger (3 g/day for 3 month) increased PON1 arylesterase activity relative to a placebo [104], an extract of Turkish oregano boosted PON1 paraoxonase and arylesterase activities relative to the control after 3 months [105], and a 2-month red sage extract incremented paraoxonase activity when compared to baseline [106]. An intervention trial with yerba mate tea (1000 mL/day) increased PON1 circulating levels relative to the control [32]. However, another intervention (1000 mL/day for 3 months) did not modify PON1 arylesterase activity [107]. Null changes were reported in three RCTs studying a polyphenol-enriched tomato juice, an oatmeal porridge enriched with sea buckthorn flavonols, and dietary doses of vitamin E (15 mg/day) [108,109,110]. Finally, regarding other HDL properties potentially related to antioxidant/anti-inflammatory potential, an increased consumption of lycopene decreased the levels of HDL-bound acute-phase cytokines, such as serum amyloid A [91].

From a total of 26 studies with antioxidants, some effect on HDL function was reported in 5 out of 11 studies with healthy individuals, and 11 out of 15 in subjects with cardiovascular risk-related pathologies. A considerable part of the effects described in subjects with pathologies employed PON methodologies.

#### 3.5.2. Antioxidant-Rich Dietary Patterns

Five human trials assessed the overall effect on HDL functions of an increase in the intake of dietary antioxidants through healthy dietary patterns (Table 7).

Two interventions in a long-term RCT (1 year) with Mediterranean diets enriched with virgin olive oil or mixed nuts were linked to increases in CEC values relative to baseline [19]. The same intervention with a Mediterranean diet enriched with virgin olive oil was also associated with decreased CETP activity relative to baseline and an improvement in an indirect indicator of LCAT function [19]. CETP activity also decreased relative to baseline in another 1-year intervention with an olive-oil-enriched Mediterranean diet [28]. A diet rich in fruits and vegetables (6 portions/day of fruit and vegetables for 8 weeks) was related to an increase in LCAT activity relative to baseline [111]. In contrast, a diet rich in vegetables, berries, and apples presented no changes in LCAT activity at 6 weeks [96].

Regarding the relationship between HDL and oxidative stress in RCTs, HDLs became more resistant to oxidation relative to baseline after the intervention with a Mediterranean diet enriched with virgin olive oil [19]. They also increased their content of carotenoids (α-carotene, β-cryptoxanthin, lutein, and lycopene) after an intervention with a diet rich in fruits and vegetables, relative to a diet lacking in plant-based foods [111]. In addition, HDL capacity to directly decrease LDL oxidation improved after the intervention with a virgin olive-oil-rich Mediterranean diet [19]. Following this same Mediterranean diet and a diet rich in fruits and vegetables also increased PON1 arylesterase activity when compared to control diets in two large RCTs [19,111]. Conversely, a 5-week vegetable-rich diet decreased PON1 activity relative to a low-vegetable diet [112]. A second trial based on a 6-week high intake of vegetables, berries, and apples decreased PON1 activity relative to baseline [96]. Antioxidant-rich diets have additionally been shown to improve the role of HDL in low-grade inflammation. In an RCT, a Mediterranean diet enriched with virgin olive oil decreased the concentrations of HDL-bound cytokines (α1-antitrypsin) relative to control interventions [21]. Finally, in two RCTs, following a Mediterranean diet rich in virgin olive oil has been observed to increase the activity of another antioxidant/antithrombotic enzyme carried by HDLs (HDL-bound phospholipase A2, also known as platelet-activating factor acetylhydrolase), as well as HDL capacity to promote the endothelial release of nitric oxide in vitro [19,21], relative to the control intervention in both cases.

Briefly, while one study with an antioxidant-rich dietary pattern was performed in healthy subjects without benefits to HDL function, four more were conducted in subjects at high cardiovascular risk, with benefits reported in all of them.

### 3.6. Ethanol and HDL Function

Moderate consumption of alcohol was analyzed in eight studies comparing the intake of wine, beer, gin, and whisky (from 15 g/day to 40 g/day) with non-alcoholic beverages (Table 8).

Ethanol intake increased CEC in healthy populations in studies comparing: red wine (30 g alcohol/day for 2 weeks) with alcohol-free wine [113]; red wine, beer, and gin (40 g alcohol/day for 3 weeks) with an equivalent volume of carbonated water [114]; white wine with grape juice (24 g alcohol/day for 3 weeks) [115]; whisky (40 g alcohol/day for 17 days) with an equivalent volume of water [116]; and beer (30 g/day alcohol –men–, 15 g/day –women– for 4 weeks) with alcohol-free beer [117]. A marginally significant increase was also observed in a trial comparing beer (1 L/day 36 g ethanol/day– for 4 weeks) with an equivalent volume of non-alcoholic beverages [118]. In contrast, alcohol intake did not modify CETP activity in two RCTs comparing red wine with alcohol-free wine (30 g alcohol/day for 2 weeks) [113] and white wine with grape juice (24 g alcohol/day for 3 weeks) [115].

Finally, alcohol intake promoted PON1 paraoxonase activity in an RCT after a 3-week consumption of 40 g alcohol/day (as red wine, beer, or gin) relative to carbonated water [119], as well as beer (40 g alcohol/day –men–, 30 g/day –women– for 3 weeks) relative to alcohol-free beer [120]. These results are consistent with the observed increase in HDL antioxidant capacity relative to baseline in LDLs after consumption of beer during a 4-week period, (30 g alcohol/day for men and 15 g/day for women) [117].

From the eight studies with ethanol intake benefits in CEC, improvements were specially observed in four studies with healthy subjects and in one with overweight/obese participants. Two studies with healthy individuals showed improvements in PON1 activity.

### 3.7. Physical Activity, Calorie Restriction, and HDL Function

Physical activity, caloric restriction, and their combinations were studied as possible promoters of HDL function in thirteen RCTs (Table 9).

Regarding CEC, the effect of a high levels of vigorous-sintensity activity (16 kcal/kg/week at 75% VO2 reserve) during a 6-month period increased radio-labeled CEC compared with two groups with different amounts of moderate-intensity physical activity and a group combining moderate-intensity exercise with a low-fat diet [20]. A 6-month high-level endurance-training intervention (with a caloric goal of 20 kcal/kg/week at 65–85% VO2) increased only the non-ABCA1 cholesterol efflux relative to a control group without exercise [20]. In the previous two studies, the same interventions evaluated with fluorescent-labeled CEC technique did not find any changes in CEC. Moreover, an RCT based on aerobic/resistance training (3 times/week for 24 weeks) also reported no effects [18]. In a similar manner, another intervention with aerobic/resistance training (4 times/week for 12 weeks) did not report changes in CEC [121]. A very low-calorie restricted diet (500 kcal/day for 6 weeks), which was accompanied by an average weight loss of 10 kg, also led to no changes in CEC [122].

The combination of physical activity and a hypocaloric diet was associated with improvements in CEC relative to usual care in a small group of obese adolescents with a diet of 1500–1800 kcal combined with supervised exercise sessions for 10 months (aerobic/resistance training 3 times/week and 2 h/day of lifestyle activities) [23]. A second trial reported improvements in CEC relative to baseline levels after a combination of aerobic training (4 times/week) and a DASH diet (with reductions of 600 kcal/day) for 12 weeks in a small group of metabolic syndrome participants [123]. However, no differences were found in a trial combining calorie-restriction counselling and aerobic exercise (mainly brisk walking 5 times per week) [124].

Regarding CETP, 6 months of aerobic training (4 times/week) did not modify its activity [22]. Nevertheless, a 6-week endurance-training intervention (3–5 times/week) decreased CETP activity relative to an untrained control group [125]. Two weight-loss interventions did not modify CETP, either in an intervention with the National Cholesterol Education Program Step I or in a 4-week intervention [126] with a very low-calorie diet (500 kcal/day for 6 weeks) [8,126]. LCAT activity and plasma concentrations were unaffected by aerobic activity (ranging from 11 weeks to 1 year) in three RCTs performed some years ago [27,127,128].

A study with of aerobic and resistance exercise programs for 12 weeks in obese women decreased serum PON1 activity relative to a control without exercise. However PON1 expression in isolated HDLs remained unaltered [121]. The same study also found no changes in the expression of isolated HDLs of the antioxidant/antithrombotic enzyme phospholipase A2 bound in HDLs [121].

Finally, regarding HDL endothelial properties, an intervention with a healthy diet combined with exercise (1500–1800 kcal/day) and aerobic/resistance training improved the HDL role in the activation of endothelial nitric oxide synthase compared to a usual-care intervention [23]. In contrast, another RCT found an aerobic and resistance exercise program for 12 weeks in obese women did not modify HDL anti-inflammatory capacity in endothelial cells [121].

In summary, there were six studies of healthy subjects with physical activity and/or calorie restriction, and in one of them, a benefit to CETP was described. From eight papers with participants presenting cardiovascular risk-related pathologies, some effect—mainly on CEC—was reported in three of them.

### 3.8. Other Lifestyle Interventions

Very few studies have evaluated dietary interventions unrelated to fats, antioxidants, ethanol, and physical activity/calorie intake (Table 10).

The effect of a prebiotic and probiotic-enriched pasta (with β-glucans 2.3 g/100 g- and Bacillus coagulans) was evaluated in a 12-week intervention. This study increased ABCG1-mediated CEC relative to a control pasta [129].

Only one RCT has evaluated the effects of the isocaloric exchange between hig- and low-glycemic-index carbohydrates on HDL function. This study found no effects on CEC [130].

Interventions with soy protein reported no changes in CEC after 6 weeks of 25–50 g soy protein and a control [131], no changes in CETP mass and LCAT activity after a 4-week, 20 g/day intake of soy protein [132], and an increase in PON1 activity in a trial of 50 g/soy protein relative to a placebo [133].

One RCT evaluated the association of carbohydrate restriction and HDL functions and found it was related to decreased LCAT activity relative to baseline but no changes in CETP function [134].

A whole dietary intervention with TLC/Step 2 diet for 32 days, a dietary pattern low in fats, and rich in vegetables, carbohydrates and fiber, did not change CETP activity [135].

Two RCTs investigated the effects of a supplement of psyllium fiber (one of them also included plant sterols) and reported a decrease in CETP activity relative to placebo pills but no changes in LCAT function [136,137].

Finally, plant sterols were also studied in two RCTs and were associated with decreases in CETP mass relative to a control after 4 weeks of margarine with phytosterols (1.68 g/day) [138] and relative to baseline values after 4 weeks of 2–3 g/day plant stanol [30]. Soy intervention did not modify LCAT activity [138].

There were 11 studies with interventions other than those previously referred to, five papers (some effect on HDL function in three) with healthy subjects, and six studies (some effect in four) with cardiovascular risk-related pathologies.

## 4. Discussion

In this systematic review, we have summarized the existing evidence regarding the effect of lifestyle changes on HDL functional traits. Short-term consumption of dietary antioxidants and alcohol was more clearly related to HDL functional improvements in subjects with cardiovascular risk and healthy individuals, respectively, especially regarding CEC and HDL antioxidant properties. Additionally, in subjects at cardiovascular risk, an effect on HDL functions was suggested after the intake of MUFAs and long-chain PUFAs, whilst an antioxidant-rich dietary pattern was able to improve HDL function in both groups, healthy individuals and subjects at high cardiovascular risk.

MUFA and long-chain PUFA intake has been analyzed in a considerable number of studies, with controversial results. Such diverse findings can be partially explained by the high heterogeneity of the study designs, for instance, the different types of fatty acids used in diets/supplements and control arms, and the wide range of doses employed. The clearest improvements in HDL functions were observed when MUFAs and PUFAs were compared to low-fat, SFA, and TFA interventions. However, the comparisons between MUFAs and PUFAs and between different types of PUFAs (linoleic acid, linolenic acid, EPA, and DHA) did not show differences regarding HDL functionalities. MUFA interventions reported increased levels of CEC, CETP, and LCAT only when compared with SFAs and TFAs. Such results are consistent with previous non-randomized studies in humans in which increases in CEC [139], CETP [140,141], and LCAT activity were also reported [142]. All long-chain PUFA studies consistently improved HDL antioxidant capacity through augmented PON1 activity. Antioxidant and anti-inflammatory capacities were also described in non-randomized trials in which doses of EPA-DHA (1.8–2 g/day) increased PON1 mass and arylesterase activity and decreased the expression/secretion of pro-inflammatory proteins, such as VCAM-1, alpha-1-antitrypsin, and complement proteins [143,144]. Long-chain fatty-acid intake also suggested improvements in CEC (when not compared to other unsaturated fatty acid interventions), accompanied by improvements in CETP and LCAT activities. Changes in CEC could be caused by increments in apolipoprotein A-I, the major apolipoprotein in HDL involved in CEC, as observed after the consumption of EPA + DHA [61,144]. On the other hand, a detrimental effect of higher doses of EPA and DHA (>3 g/day) on LCAT and HDL inflammatory indices, with no changes in CEC, has been proposed. Hypotheses describing the potential beneficial effects of MUFAs and PUFAs on HDL function are diverse. A first explanation lies in the capacity of these fatty acids to increase HDL particle fluidity. HDLs, which are more fluid, are thought to have a greater capacity to adapt to the shape of cholesterol transporters in cells and allow for the export of cholesterol excess [139,145,146]. Second, omega-3 PUFAs are known ligands for peroxisome-proliferator-activated receptor α (PPARα). This cell receptor, the activation of which leads to an increased production of apolipoprotein A-I (the HDL main active protein), may improve HDL function beyond an increase in HDL-C levels [147,148]. An increase in apolipoprotein A-I levels could also mediate a greater stability and antioxidant function of PON1 [149] beyond the intrinsic potential capacity of PUFAs to increase hepatic synthesis and its release [150]. Third, omega-3 PUFAs are able to decrease low-grade inflammation. This may lead to a reduction in the circulating levels of cytokines and acute-phase proteins, thus favoring a lower binding of these molecules to HDL particles [151]. Moreover, most of these interventions are potentially rich in dietary antioxidants, which may additionally contribute to the effects observed [152]. Finally, we cannot exclude the fact that the decrease in triglyceride concentrations due to PUFAs may play a role in explaining some of the benefits on HDL cholesterol metabolism. PUFAs can reduce hepatic synthesis of triglycerides and very low-density lipoproteins (VLDL), activating PPAR receptors [153]. Considering the close relationship between both lipids, decreases in triglycerides and VLDL could decrease CETP enzymatic activity and improve HDL cholesterol metabolism [154].

Saturated and trans fats presented opposite effects to MUFAs and PUFAs, with lower CEC and LCAT and increased CETP. Nevertheless, an article that studied different sources of saturated fats (from cheese and butter) described contraposed effects in CEC [33]. These findings could be due to the fact that the toxic effect of long-chain SFAs, such as palmitic acid [155], is not present in shorter chain species [156,157]. A few studies have also suggested a diminished HDL profile (worse PON1 and CETP activities) after TFAs when compared with SFA-rich diets [48,68,72]. Impairment in HDL functions is probably secondary to the increment of total cholesterol and fractions induced by SFAs and TFAs [155,158].

Dietary cholesterol intake promotes increases in CEC, CETP, and LCAT activities. Augmented intake may increment the pool of cholesterol in circulation (in both LDLs and HDLs), which could be related to increased cholesterol levels in the organism, consequently leading to a greater uptake of cholesterol from peripheral cells (CEC) and a greater necessity to metabolize cholesterol by CETP and LCAT in oder to transfer it back to the liver. However, increases in CEC caused by cholesterol intake could not reflect a better atheroprotective capacity of HDL. Cholesterol intake showed increases in large HDL particles and HDL diameter [82,86], which are associated with greater cardiovascular risk [6]. In addition, HDLs may be more oxidized and contain higher levels of acute-phase proteins, such as SAA, suggesting increased levels of oxidative stress and low-grade inflammation.

Under chronic oxidative stress and inflammation, HDL lipoprotein loses its atheroprotective capacity and becomes dysfunctional [11]. Thus, antioxidant-rich dietary patterns are promising interventions for the preservation of HDL atheroprotective capacity. As described in this review, olive oil enriched with phenolic compounds, anthocyanins, carotene extracts, and supplements is clearly associated with improvements in HDL functionalities. The Mediterranean diet, which includes all the previously mentioned beneficial antioxidants, presents a positive effect on a wide battery of functions (CEC, antioxidant, anti-inflammatory, and endothelial protection capacity of HDLs) [21,28,89]. In contrast, the few interventions that only increased dietary vegetable content failed to improve CETP, LCAT, and PON1 activities of HDLs [96,112]. It is possible changes in vegetable quantity without other beneficial compounds (such as olive oil rich in polyphenols, nuts, and other sources of omega-3 fatty acids) are not enough to change HDL functions in a whole dietary pattern. Several intertwined molecular mechanisms have been suggested as hypothetical explanations. First, a number of HDL proteins involved in reverse cholesterol transport (apolipoprotein A-I, LCAT) and antioxidant capacity (PON1) have been described as having their functional capacity decreased if they become oxidized [159,160,161]. Thus, antioxidants may keep these proteins non-oxidized and functional. Second, oxidized lipids are also known to become less fluid. Therefore, antioxidants may enhance HDL lipid fluidity [162]. Third, some phenolic compounds may induce a slight boost of AMP-activated protein kinase (AMPK), a cellular metabolic regulator capable of activating PPARα and the subsequent synthesis of apolipoprotein A-I [152,163]. Fourth, antioxidant-rich HDLs are hypothesized to employ the antioxidant compounds they carry to counteract the oxidation of other lipids. Fifth, regarding HDL anti-inflammatory potential, the decrease in reactive oxygen species due to dietary antioxidants may moderate the activation of the nuclear factor kappa beta (NF-κβ), a pivotal regulator of inflammatory responses [164]. This, in turn, can reduce the concentrations of cytokine/acute-phase proteins that bind to HDLs. Some particular antioxidants, such as flavonoids, have been reported to be able to directly downregulate NF-κβ activation [165]. In addition, several phenolic compounds are also able to promote an AMPK-mediated decrease in the production of low-grade inflammation signals [166,167]. Finally, the capacity of some phenolic antioxidants to decrease hepatic liver synthesis through an AMPK-dependent mechanism could also lead to a decrease in circulating levels of triglycerides [163]. This could be linked to an enhancement in HDL cholesterol metabolism due to the close relationship between HDL-C and triglycerides [154].

Alcohol consumption is clearly associated with higher levels of HDL-C, accompanied by increments of apolipoprotein A-I [168]. In the same manner, moderate ethanol consumption has presented consistent increases in CEC and PON1 activity. Although two RCTs did not demonstrate changes in CETP, some non-randomized trials have reported improvements in CETP and LCAT activities [169,170] and detrimental effects in CETP after alcohol withdrawal [171,172]. The main hypothesis to explain the potentially beneficial effects of alcohol on HDL metabolism is related to the transient increase in acetate, the main ethanol metabolite after its ingestion [173]. Acetate is a short-chain fatty acid known to be capable of decreasing lipolysis [174], which lowers levels of non-esterified fatty acids released into circulation. These fatty acids are physiologically transformed into triglycerides in the liver and subsequently packed in VLDLs. A transient decrease in VLDLs is therefore expected after alcohol intake [173]. VLDLs are the main destination of cholesterol esters collected by HDL particles in CEC. Thus, if transiently low VLDL levels are present, CETP activity is halted, and HDL-C concentrations increase due to greater cholesterol content per HDL particle (larger HDLs) [60,175]. Finally, an increment in CEC directed to these large HDLs, mediated by specific cholesterol transporters, such as ABCG1 and SR-BI, could be expected, and a subsequent esterification of the collected cholesterol by LCAT takes place [173]. In addition, the promotion in PON1 antioxidant function could be partially justified by the high content of dietary antioxidants in some alcoholic beverages [176,177].

The effect of physical activity on HDL function is still unclear. Regarding CEC, only large amounts of vigorous exercise were associated with improvements [20]. Moderate-intensity activities did not show any changes in CEC, CETP, or LCAT activities, suggesting a possible dose-dependent relationship. Further evidence was found in non-randomized trials, which also reported the absence of effect of moderate physical activity [128,178]. In addition, a very low calorific diet changed neither CEC nor CETP activity (Talbot et al., 2018). In fact, non-randomized studies even found negative effects on CEC [179,180]. Nevertheless, interventions combining physical activity with modest calorie restriction suggested increases in CEC and decreases in CETP activity [23,123] in non-randomized trials [181,182]. Physical activity and/or calorie restriction are known to promote the activation of AMPK [183], a mechanism that seems essential to the hypothetical explanation of the molecular effects of these two lifestyle modifications on HDL function [184]. First, as already mentioned, AMPK is capable of stimulating PPARα, a transcription factor that promotes the hepatic synthesis of apolipoprotein A-I and the release of cholesterol transporters in peripheral cells, such as macrophages, leading to increases in HDL-C circulating levels and potential improvements in CEC values [148]. In parallel, the beneficial decrease in CETP activity could be secondary to the reduction in triglyceride plasma levels due to the ability of AMPK to lower the production of triglyceride synthesis enzymes in the liver [185]. Second, AMPK simulation has also been linked to a greater production of several antioxidant enzymes [186,187,188]. This antioxidant protection could be related to a stronger preservation of HDL proteins, such as apolipoprotein A-I and PON1, in a non-oxidized, functional state, boosting CEC and HDL antioxidant capacities [159,161]. Finally, AMPK stimulation may counteract low-grade inflammation, which, in turn, may be related to decreased levels of cytokines and acute-phase proteins bound to HDLs and a reduction in the pro-inflammatory potential of the lipoprotein [166,167].

This systematic review has some limitations to be considered. First, the high percentage of studies classified as unclear risk could conceal the presence of bias, leading to questions of their quality. Second, there is a marked heterogeneity in the laboratory procedures used to evaluate HDL functions. For example, to evaluate CEC, the analyzed studies employed six different types of cell cultures (J774, THP-1, Fu5AH, CHO, RAW264.7, and PBMCs cells) and three different types of labeled cholesterol (2 radio-labeled and 1 fluorescent). Some of these assays could hinder standardization and measurement precision, and it remains to be proven whether they constitute adequate surrogate endpoints. Third, there was heterogeneity in the lifestyle modifications and trial designs. Fourth, most studies did not specify whether the LCAT activity was beta or alpha, and consequently, whether the activity is specifically linked to HDL. Finally, most of the studies evaluated were short-term interventions with modest sample sizes. For all these reasons, results need to be interpreted with caution.

As far as the strengths of this review, we have reviewed many publications, including an exhaustively wide range of studies of both HDL functional abilities and lifestyle interventions.

## 5. Conclusions

Given that healthy diet and lifestyle are consistently related to decreased cardiovascular disease risk, their link to improved HDL function has been widely investigated in human trials. In brief, beyond an improvement in HDL-C levels, mainly in short-term clinical trials, the consumption of MUFAs, PUFAs (particularly long-chain, omega-3 MUFAs and PUFAs in fish), and dietary antioxidants, such as phenolic compounds (in dietary or near-dietary doses), showed benefits in HDL functionality, mainly in subjects with cardiovascular risk factors. In this regard, antioxidant-rich dietary patterns were able to improve HDL function in both healthy individuals and subjects at high cardiovascular risk. In addition, reverse cholesterol transport with ethanol at moderate quantities, in studies mainly performed with healthy individuals, was able to enhance CEC. Finally, cholesterol dietary interventions with eggs increased circulant cholesterol fractions (total and HDL cholesterol) and, in concordance, increase CEC, CETP, and LCAT activity.

Such findings suggest the capacity of dietary and lifestyle modifications to modulate cardiovascular risk factors. Nevertheless, given the marked heterogeneity in study design and procedures used to assess HDL functions, more homogeneous, large-scale, long-term, randomized, controlled trials are required to confirm these results. Moreover, such trials should be performed over different periods, in varying populations, and with individuals presenting diverse pathologies.

## Figures and Tables

**Figure 1 jcm-10-05897-f001:**
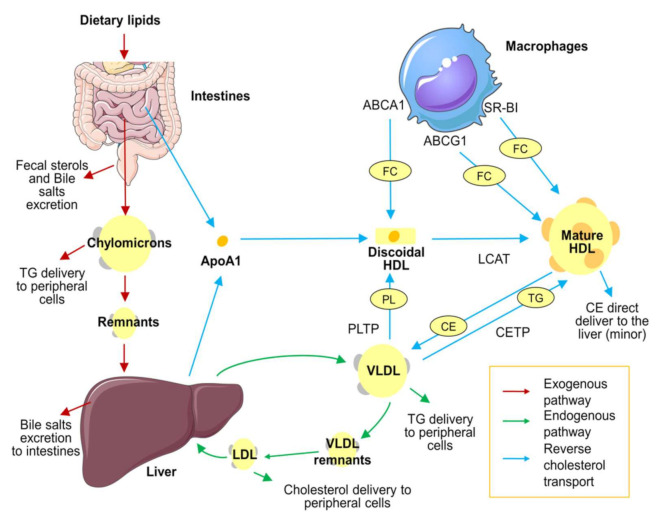
Lipoprotein metabolism overview. Lipid distribution in the body occurs in three different pathways. First, the exogenous pathway (red arrows): The absorption of dietary triglycerides, free cholesterol, and cholesteryl esters occurs in the small intestine. In enterocytes, dietary lipids are packed in chylomicrons and diffused to the bloodstream. Chylomicrons diffuse triglycerides to peripheral cells, and their remnants are therefore cleared in the liver. Second, the endogenous pathway (green arrows): Triglycerides and cholesterol synthetized in the liver are recirculated in the bloodstream packed in VLDL. VLDL transports triglycerides to peripheral cells. VLDL remnants are transported to the liver, where remaining triglycerides are removed by hepatic lipase action and become LDL. LDL transports cholesterol to peripheral tissues and is eventually cleared by the liver. Finally, HDLs are responsible for reverse cholesterol transport (blue arrows): ApoA1 is synthetized in the liver and enterocytes and released as a lipid-free monomer. Then it incorporates phospholipids by action of PLTP from VLDL. Lipid-free ApoA1 is able to collect free cholesterol of peripheral cells, such macrophages, through the ABCA1 receptor. The accumulation of phospholipids and free cholesterol results in the formation of discoidal HDL. Free cholesterol is transformed to cholesteryl esters and continuously internalized in the HDL core by LCAT enzymes, forming the mature form of HDL. Mature HDL continues to pick up cholesterol through ABCG1 and SR-BI receptors. Finally, the accumulated cholesterol can be transported back to the liver, mainly in an indirect way (exchanging cholesteryl esters for triglycerides with VLDL through CETP activity) or, in a minor proportion, directly through hepatic receptors. The figure was produced using Servier Medical Art (http://smart.servier.com/ accessed on 6 December 2021). ABCA1: ATP-binding cassette transporter A1. ABCG1: ATP-binding cassette transporter G1. ApoA1: Apolipoprotein A1. CE: cholesterol esters. CETP: cholesteryl ester transfer protein. FC: free cholesterol. HDL: high-density lipoprotein. LCAT: lecithin cholesterol acyltransferase. LDL: low-density lipoprotein. PL: phospholipid. PLTP: phospholipid transfer protein. TG: triglycerides. VLDL: very low-density lipoprotein.

**Figure 2 jcm-10-05897-f002:**
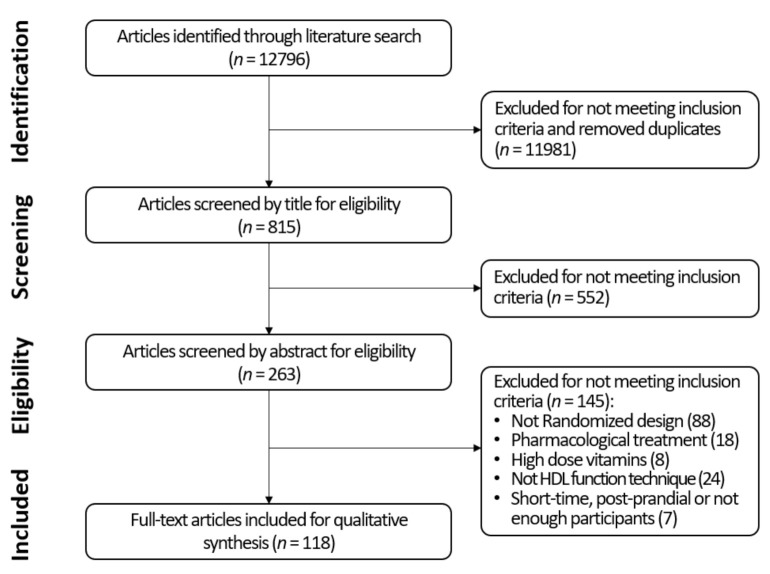
Flow diagram of study selection.

**Figure 3 jcm-10-05897-f003:**
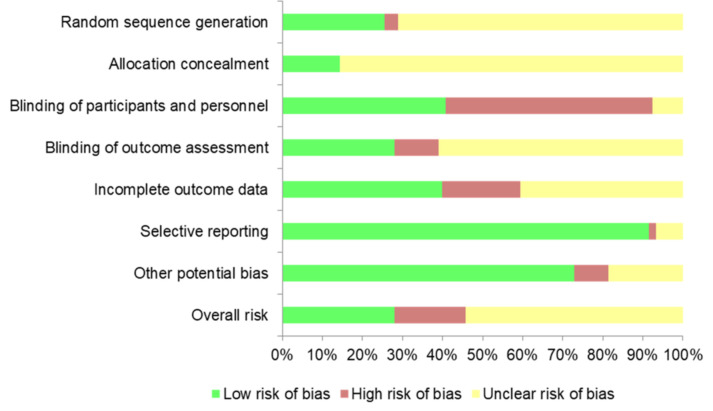
Overall risk of bias across studies.

**Table 1 jcm-10-05897-t001:** Studies with monounsaturated fatty acids (MUFA): oleic-acid-rich oil.

First Author, Location (Year)	Study Participants	Intervention	HDL Function Analyzed	Results
Andraski, USA (2019) [39]	12 overweight participants (5 men and 9 women) Mean age: 42.16 years	32-day crossover diets with: (1) High-MUFA diet (23% of total calorie content). (2) High-carbohydrate diet (65% of total calorie content).	Plasma LCAT activity	No effect.
Baudet MF, France (1988) [37]	20 healthy women Mean age ± SD: 39 ± 9 years	6-week crossover intervention diets with 15.6% of total calories from: (1) Low-erucic-acid rapeseed oil. (2) Sunflower oil. (3) Peanut oil. (4) Milk fats (butter or cream).	Serum LCAT activity	Increase in LCAT activity in peanut group relative to milk-fat diets or low-erucic-acid rapeseed oil.
Brassard D Canada, (2018) [33]	77 abdominal obesity patients (21 men and 25 women) Mean age: 41.4 ± 14.2 years	5-week crossover intervention with: (1) Cheese-rich diet. (2) Butter-rich diet. (3) Olive-oil-rich diet. (4) Corn-oil-rich diet. (5) Carbohydrate-rich diet.	^3^H CEC in J774 cells in ABDP samples	Increase in olive oil intervention relative to cheese and carbohydrate diet.
Lagrost L, France (1999) [35]	32 healthy participants (14 men and 18 women) Age range: 20–60 years	6-week crossover diets enriched with: (1) Palmitic acid (45% total fats). (2) Lauric acid (44% total fats). (3) Oleic acid (62% total fats).	Serum CETP activity and mass	Decreased CETP activity and mass in oleic-acid diet relative to palmitic-acid and lauric-acid diets.
Liu X, Canada/USA (2018) [29]	101 metabolic syndrome participants (50 men and 51 women) Mean age: 49.5 years	4-week crossover design with 5 isocaloric diets supplemented with 60 g of: (1) Canola oil. (2) Canola oil high-oleic-acid content. (3) Canola oil high in DHA and oleic acid. (4) Corn oil combined with safflower oil. (5) Safflower oil combined with flax oil.	3-NBD CEC in THP-1 in serum samples	Increase by 39.1% in the canola-oil group, 33.6% in canola oil rich in oleic acid, and 55.3% in canola oil rich in DHA and oleic acid, relative to baseline levels.
Singer P, Germany (1990) [38]	40 men with mild essential hypertension	2-week parallel diets with 60 mL/day of: (1) Olive oil. (2) Sunflower oil. (3) Linseed oil.	LCAT activity	No effect.
Stirban A, Germany (2014) [40]	34 participants with type 2 diabetes Mean age: 56.8 years	6-week parallel intervention with: (1) 2 g of EPA and DHA. (2) Olive-oil placebo.	Serum Paraoxonase-1 activity	No effect.
Solà R, Spain (1997) [34]	22 healthy men Mean age ± SD: 49.7± 0.6 years	8-week crossover intervention with isocaloric diets with 15.6% of: (1) Sunflower oil rich in oleic acid. (2) Sunflower oil rich in linoleic acid.	^3^H CEC in primary macrophage cells in isolated HDL3	No effect.
			HDL3 oxidation status by TBARS assay	Decrease in malondialdehyde production in oleic-rich diet compared to linoleic-rich diet.
Vega-López S, USA (2006) [36]	15 participants with high levels of LDL cholesterol (5 men and 15 women) Mean age ± SD: 63.9 ± 5.7 years	5-week crossover intervention with four diets with 20% fat content provided by: (1) Partially hydrogenated soybean oil (13% TFAs). (2) Soybean oil (44% PUFAs). (3) Palm oil (50% SFAs). (4) Canola oil (49% MUFAs).	Plasma CETP activity	No effect.

SFA: saturated fatty acids. TFA: trans fatty acids. MUFA: monounsaturated fatty acids. PUFA: polyunsaturated fatty acids. DHA: docosahexaenoic acid. EPA: eicosapentaenoic acid. CEC: cholesterol efflux capacity. CETP: cholesteryl ester transfer protein. LCAT: lecithin–cholesterol acyltransferase. ABDP: apolipoprotein B-depleted plasma TBARS: thiobarbituric-acid-reactive substance.

**Table 2 jcm-10-05897-t002:** Studies with polyunsaturated fatty acids (PUFA): vegetable oils and nuts.

First Author, Location (Year)	Study Participants	Intervention	HDL Function Analyzed	Results
Abbey M, Australia (1990) [53]	33 hypercholesterolemic men Mean age ± SD: 47.4 ± 2.5 years	6-week parallel intervention with supplements of: (1) 14 g/day linoleic acid (safflower oil). (2) 9 g α-linolenic acid (linseed oil). (3) 3.8 g of n-3 FA (fish oil).	Plasma LCAT activity	No effect.
Baudet MF, France (1988) [37]	20 healthy women Mean age ± SD: 39 ± 9 years	6-week crossover intervention diets with 15.6% total calories from: (1) Low-erucic-acid rapeseed oil. (2) Sunflower oil. (3) Peanut oil. (4) Milk fats (butter or cream).	Serum LCAT activity	Increase in LCAT activity in sunflower oil group relative to milk fats diets or low erucic rapeseed oil.
Berryman CE, USA (2017) [41]	48 participants with high LDL cholesterol (22 men and 26 women) Mean age ± SD: 50 ± 9 years	6-week crossover intervention with: (1) 43 g almonds/day. (2) Control diet (isocaloric cholesterol-lowering diet (without almonds)).	^3^H CEC in J774 cells in ABDS samples	Increase in non-ABCA1 CEC relative to control diet.
Buonacorso V, Brazil (2007) [44]	30 healthy participants (9 men and 21 women) Mean age ± SD: 35.3 ± 10 years	4-week parallel intervention with diets enriched with: (1) TFAs (8.3% total energy). (2) PUFAs (14.6% total energy). (3) SFAs (13.2% total fats).	^3^H CEC in primary macrophages cells in isolated HDL3 and HDL2	No effect.
Canales A, Spain (2011) [55]	22 participants at high cardiovascular risk (12 men and 10 women) Mean age: 54.8 years	5-week crossover intervention with: (1) 300 g/week of walnut-enriched meat (20% walnut paste). (2) Control low-fat diet.	Serum Paraoxonase 1 activity	Increased PON1 activity inn walnut meat group relative to control diet.
Canales A, Spain (2007) [56]	22 participants at high cardiovascular risk (12 men and 10 women) Mean age: 54.8 ± 8.3 years	5-week crossover intervention with: (1) 600 g/week of walnut-enriched meat (20% walnut paste). (2) Control low-fat diet.	Serum Paraoxonase 1 activity	Increased PON1 activity inn walnut meat group relative to baseline.
Chung BH, USA (2004) [51]	16 healthy participants (8 men and 8 women) Mean age men ± SD: 35.3 ± 4.5 years Mean age women ± SD: 51.9 ± 6.6 years	16-day crossover diet: (1) PUFA-rich diet. (2) SFA-rich diet.	Plasma CETP mass	No effect.
Cox C, New Zeland (1995) [47]	28 hypercholesterlemic participants (13 men and 15 women) Age range: 26–64 years	6-week crossover isocaloric diets: (1) Safflower-oil diet. (10% energy from PUFAs) (2) Coconut-oil diet (20% energy from SFAs). (3) Butter diet (20% energy from SFAs).	Cholesteryl ester transfer activity (CETA)	Decreased CETA activity in Safflower oil group relative to butter intervention.
De Souza, Brazil (2018) [49]	46 overweight or obese women Age range: 20–59 years	8-week parallel isocaloric diets: (1) 20 g/day of baru almonds. (2) Control diet with 800 mg maltodextrin supplement.	Plasma CETP mass	Decrease in baru-almond diet relative to control diet.
Gebauer SK, USA (2008) [50]	28 hypercholesterolemic patients (10 men and 18 women) Mean age ± SD: 48 ± 1.5 years	4-week crossover intervention with: (1) 10% energy from pistachios. (2) 20% from pistachios. (3) Control low-fat diet.	Serum CETP mass	No effect.
Holligan SD, USA (2014) [42]	28 participants with high LDL cholesterol (10 men and 18 women) Mean age ± SD: 48.0 ± 1.5 years	4-week crossover intervention with: (1) 10% energy from pistachios. (2) 20% from pistachios. (3) Control low-fat diet.	^3^H CEC in J774 cells in ABDS samples	Increase ABCA1 CEC in 20% pistachio diet, relative to 10% pistachio diet, in participants with low CRP levels.
Kawakami Y, Japan (2015) [46]	26 healthy men Mean age ± SD: 44.5 ± 3.1 years	12-week crossover diet interventions: (1) 10 g of flaxseed oil (5.49 g of α-linolenic acid). (2) 10 g of corn oil (0.09 g of α-linolenic acid).	CETP mass	Flaxseed oil decreased CETP mass compared to corn oil.
Kralova-Lesna I, Czech Republic (2008) [45]	14 healthy men Age range: 18–55 years	4-week crossover intervention with two diets with diets containing 40% from fats: (1) High-SFA diet (52% SFA). (2) High-PUFA diet (41% PUFA).	^14^C CEC in THP-1 cells in serum	No effect.
Liu X, Canada/USA (2018) [29]	101 metabolic syndrome participants (50 males and 51 females) Mean age: 49.5 years	4-week crossover design with 5 isocaloric diets with 60 g of: (1) Canola oil. (2) Canola oil with high oleic-acid content. (3) Canola oil high in DHA and oleic acid. (4) Corn oil combined with safflower oil. (5) Safflower oil combined with flax oil.	3-NBD CEC in THP-1 in serum samples	Increase of 49.2% in corn oil + safflower oil and 50.7% in safflower oil combined with flax oil, relative to baseline levels.
Lottenberg AM, Canada (1996) [52]	19 hypercholesterolemic women Mean age ± SD: 51.3 ± 12.7 years	3-week crossover diet: (1) High-SFA diet (45% total fat from SFA oil). (2) High-PUFA diet (50% total fat from PUFA oil).	Plasma CETP activity and mass	No effect.
Pfeuffer M, Germany (2011) [57]	85 obese men Age range: 45–68 years	4-week intervention with supplements of: (1) 4.5 g/day conjugated linoleic acid. (2) Safflower oil. (3) Heated safflower oil. (4) Control olive oil.	Paraoxonase 1 and arylesterase activity	Increase in arylesterase activity in both safflower oil interventions compared to a conjugated linoleic acid group.
Sánchez-Muniz FJ, Spain (2012) [54]	22 participants at high cardiovascular risk (12 men and 10 women) Mean age: 54.8 years	4 to 6 week crossover intervention with: (1) 750 g/week of walnut-enriched meat (20% walnut paste). (2) Control low-fat diet.	Paraoxonase 1 activity	Increased PON1 activity in walnut-enriched meat group relative to control diet.
Singer P, Germany (1990) [38]	40 males with mild essential hypertension	Parallel diets with 60 mL/day of: (1) Olive oil. (2) Sunflower oil. (3) Linseed oils.	LCAT activity	Decrease in LCAT activity relative to baseline intervention.
Solà R, Spain (1997) [34]	22 healthy men Mean age ± SD: 49.7 ± 0.6 years	8-week crossover intervention with isocaloric diets with 15.6% of: (1) Sunflower oil rich in oleic acid. (2) Sunflower oil rich in linoleic acid.	^3^H CEC in primary macrophage cells in isolated HDL3	No effect.
			Oxidation status of HDL3 by TBARS assay	Increase in malondialdehyde production in linoleic-rich diet compared to oleic-rich diet.
Tindall AM, USA (2020) [43]	34 participants at high cardiovascular risk (21 men and 13 women) Mean age ± SD: 44 ± 10 years	6-week crossover diet interventions: (1) Walnut diet (57–99 g/day walnut, 16% PUFAs). (2) Walnut fatty-acid–matched diet (linolenic acid matched (16% PUFAs)). (3) High oleic diet (12% MUFAs).	CEC in J774 cells in ABDS	No effect.
Van Tol A, The Netherlands (1995) [48]	55 healthy participants (25 men and 30 women) Age range:19–49 years	17-day parallel isocaloric diets with 8% of energy from: (1) Linoleic acid. (2) Stearic acid. (3) Trans fatty acid.	ABDP CETP activity	Decrease in linoleic-rich diet relative to trans-fatty-acid diet.

SFA: saturated fatty acids. TFA: trans fatty acids. MUFA: monounsaturated fatty acids. PUFA: polyunsaturated fatty acids. LTP: lipid transfer protein. PON1: paraoxonase-1 activity. DHA: docosahexaenoic acid. EPA: eicosapentaenoic acid. CEC: cholesterol efflux capacity. CETP: cholesteryl ester transfer protein. CETA: cholesteryl ester transfer activity. LCAT: lecithin–cholesterol acyltransferase. PON1: paraoxonase-1. ABDP: apolipoprotein B-depleted plasma. TBARS: thiobarbituric-acid-reactive substance.

**Table 3 jcm-10-05897-t003:** Studies with polyunsaturated fatty acids (PUFA): fish, eicosapentaenoic and docosahexaenoic acids (EPA, DHA).

First Author, Location (Year)	Study Participants	Intervention	HDL Function Analyzed	Results
Abbey M, Australia (1990) [53]	33 hypercholesterolemic men Mean age (SD): 47.4 ± 2.5 years	6-week supplement with: (1) 14 g/day linoleic acid (safflower oil). (2) 9 g α-linolenic acid (linseed oil). (3) 3.8 g fish oil (EPA + DHA).	Plasma LCAT activity	Decrease of 21% in fish oil relative to baseline.
Calabresi, Italy (2004) [54]	14 participants with familial hypercholesterolemia	4-week crossover design with capsules of: (1) 4 g (EPA + DHA) and 4 mg α-tocopherol. (2) Placebo.	^3^H CEC in Fu5AH cells in plasma	No effect.
			Plasma CETP mass	No effect.
			Plasma paraoxonase-1 mass	Higher PON1 mass in omega 3 relative to placebo.
Ghorbanihaghjo A, Iran (2012) [57]	83 women with rheumatoid arthritis Mean age (SD): 50 (18–74) years	12-week parallel intervention with capsules of: (1) Fish oil (1 g/day). (2) Placebo.	Paraoxonase-1 mass in HDL	Higher PON1 content in omega 3 group compared to phytosterol-supplemented group.
Golzari MH, Iran (2017) [63]	36 patients with type 2 diabetes Age range: 35–50 years	8-weeks parallel intervention with capsules of: (1) Fish oil (EPA 2 g/day). (2) Placebo.	Serum paraoxonase-1 activity	Increase in EPA group compared to placebo.
Lambert C, Spain (2017) [61]	32 overweight or obese participants 13 men and 19 women Mean age (SD): 50.5 ± 1.6 years	4-week crossover design with: (1) Omega 3-supplemented milk (131.25 mg EPA + 243.75 mg DHA/250 mL). (2) Phytosterol-supplemented milk (1.6 g of plant sterols/250 mL).	Serum LCAT mass	No effect.
Liu X, Canada/USA (2018) [29]	101 metabolic syndrome participants 50 males and 51 females Mean age: 49.5	4-week crossover design with five isocaloric diets with 60 g of: (1) Canola oil. (2) Canola oil with high oleic-acid content. (3) Canola oil high in DHA and oleic acid. (4) Corn oil combined with safflower oil. (5) Safflower oil combined with flax oil.	3-NBD CEC in THP-1 in serum samples	Increase of 55.3% in canola oil rich in DHA and oleic acid relative to baseline levels.
Manninen, Finland (2019) [58]	79 participants with impaired glucose metabolism Mean age (SD): 58.9 ± 6.5 years	12-week intervention with four parallel isocaloric diets with: (1) 27 g/day camelina oil (10 g ALA). (2) Fatty fish (1 g/day DHA + EPA). (3) Lean fish. (4) Control-diet group.	^3^H CEC in primary macrophage cells in isolated HDL	No effect.
Pownall HJ, USA (1999) [60]	56 participants (40 with hypertriglyceridemia and 16 healthy) 24 men and 17 women Mean age (SD): 51.4 ± 1.9 years	Two 6-week parallel interventions with capsules of: (1) 4 g fish oil (EPA + DHA) with 4 mg α-tocopherol. (2) Placebo.	Serum cholesteryl ester transfer activity (CETA)	Decrease of 20% in fish-oil group relative to baseline levels.
Shidfar F, Iran (2016) [64]	76 women with iron deficiency Mean age (SD): 33.03 ± 8.73 years	12-week parallel intervention with capsules of: (1) 500 mg of DHA+ iron supplement (2) Placebo + iron supplement	Serum paraoxonase-1 mass	No effects.
Stirban A, Germany (2014) [40]	34 patients with type 2 diabetes	6-week parallel intervention with capsules of: (1) 2 g EPA + DHA supplement (2) Placebo supplement of olive oil.	Serum paraoxonase-1 activity	No effects.
Wurm R, Austria (2018) [65]	40 advanced heart failure participants (34 men and 6 women)	12-week parallel intervention with capsules of: (1) 1 g EPA + DHA. (2) 4 g EPA + DHA. (3) Placebo.	HDL oxidative/inflammatory index (HOII) in ABDP	Increase in HOII after 4 g fish oil per day, relative to 1 g and placebo group.

LTP: lipid transfer protein. ALA: alpha-linolenic acid. DHA: docosahexaenoic acid. EPA: eicosapentaenoic acid. CEC: cholesterol efflux capacity. CETP: cholesteryl ester transfer protein. CETA: cholesteryl ester transfer activity. LCAT: lecithin–cholesterol acyltransferase. ABDP: apolipoprotein B-depleted plasma. ABDS: apolipoprotein B-depleted serum. PON1: paraoxonase-1. TBARS: thiobarbituric-acid-reactive substance. HOII: HDL oxidative/inflammatory index.

**Table 4 jcm-10-05897-t004:** Studies with saturated (SFA) and trans fatty acids (TFA).

First Author, Location (Year)	Study Participants	Intervention	HDL Function Analyzed	Results
Baudet MF, France (1988) [37]	20 healthy women Mean age ± SD: 39 ± 9 years	6-week crossover intervention diets with 15.6% of total calories from: (1) Low-erucic-acid rapeseed oil. (2) Sunflower oil. (3) Peanut oil. (4) Milk fats (butter or cream).	Serum LCAT activity	Decrease in LCAT activity of milk fats relative to peanut oil group.
Brassard D, Canada (2018) [33]	77 abdominal obesity patients (21 men and 25 women) Mean age ± SD: 41.4 ± 14.2 years	5-week crossover intervention with: (1) Cheese-rich diet. (2) Butter-rich diet. (3) Olive-oil-rich diet. (4) Corn oil. (5) Carbohydrate-rich diet.	^3^H CEC in J774 cells in ABDP samples	Increase in butter intervention compared to cheese or carbohydrate diets.
Buonacorso V, Brazil (2007) [44]	30 healthy participants (9 men and 21 women) Mean age ± SD: 35.3 ± 10 years	4-week parallel intervention with diets enriched with: (1) TFAs (8.3% total energy). (2) PUFAs (14.6% total energy). (3) SFAs (13.2% total fats).	^3^H CEC in primary macrophages cells in isolated HDL3 and HDL2	No effect.
Chardigny JM, France (2008)	40 healthy participants (19 men and 21 women) Mean age ± SD: 27.6 ± 7.1 years	3-week crossover intervention with food items containing: (1) TFAs from natural sources. (2) TFAs from industrial sources.	Plasma CETP activity	No effect.
Chung BH, USA (2004)	16 healthy participants (8 men and 8 women) Mean age men ± SD: 35.3 ± 4.5 years Mean age women ± SD: 51.9 ± 6.6 years	16-day crossover diets: (1) PUFA-rich diet. (2) SFA-rich diet.	Plasma CETP mass	No effect.
Cox C, New Zeland (1995) [47]	28 hypercholesteremic participants (13 men and 15 women) Age range: 26–64 years	6-week crossover isocaloric diets: (1) Safflower-oil diet. (10% energy from PUFAs) (2) Coconut-oil diet (20% energy from SFA). (3) Butter diet (20% energy from SFA).	Cholesteryl ester transfer activity (CETA)	Increase in CETA activity in butter intervention group relative to safflower intervention.
de Roos NM, Netherlands (2002) [72]	32 healthy participants (11 men and 21 women) Age range: 18–69 years	4-week crossover intervention diets with: (1) SFA-rich margarine (0.3% TFAs). (2) Trans FA-rich margarine (9.3% TFAs).	Paraoxonase-1 activity	TFA group decreased by 6% PON1 activity compared to SFA group.
Lagrost L, France (1999) [35]	32 healthy participants (14 men and 18 women) Age range:20–60 years	6-week crossover diets enriched with: (1) Palmitic acid (45% total FA). (2) Lauric acid (44% total FA). (3) Oleic acid (62% total FA).	Serum CETP activity and mass	Higher CETP activity and mass in palmitic and lauric acid groups relative to oleic acid group.
Lichtenstein AH, USA (2001) [68]	36 participants with high LDL cholesterol (18 men and 18 women) Mean age ± SD: 63 ± 6 years	5-week crossover interventions with 20% calories from: (1) Semiliquid margarine. (2) Stick margarine. (3) Butter.	Plasma CETP activity	Increase in CETP activity in stick margarine group relative to butter or semiliquid margarine.
Lottenberg AM, Canada (1996) [52]	19 hypercholesterolemic women Mean age ± SD: 51.3 ± 12.7 years	3-week crossover diets: (1) High-SFA diet (45% total fat from SFA oil) (2) High-PUFA diet (50% total fat from PUFA oil).	Plasma CETP activity and mass	No effect.
Matthan NR, USA (2001) [69]	14 women with high LDL cholesterol Age range: 65–71 years	5-week crossover interventions with 20% calories from: (1) Soybean oil (0.6% TFAs). (2) Low-trans squeeze margerines (0.9% TFA). (3) Medium-trans tub margerines (3.3% TFAs). (4) High-trans stick (6.7% TFAs) margarines.	Plasma CETP activity	No effect.
Schwab US, Finland (1995) [66]	15 healthy women Age range: 19–34 years	5-week parallel diets with 36% fats from: (1) Palmitic-enriched diet (22–33 g palm oil). (2) Lauric-acid-enriched diet (16–26 g coconut oil).	Plasma CETP activity	Increase in CETP activity in lauric acid group relative to baseline.
Tholstrup T, Denmark (2004) [67]	17 healthy men Mean age ± SD: 23.4 ± 2.2 years	3-week crossover interventions with 70 g fats containing: (1) Medium-chain fatty acids (65% caprylic acid and 33% capric acid). (2) High-oleic-acid sunflower oil.	Plasma CETP activity	No effect
Van Tol A, the Netherlands (1995) [48]	55 healthy participants (25 men and 30 women) Age range:19–49 years	17-day isocaloric parallel diets with 8% energy from: (1) Linoleic acid. (2) Stearic acid. (3) Trans fatty acid.	ABDP CETP activity	Increase in trans fatty acids diet relative to linoleic rich diet.
Vega-López S, USA (2006) [36]	15 participants with high levels of LDL cholesterol (5 men and 15 women) Mean age ± SD: 63.9 ± 5.7 years	5-week crossover interventions with four diets with 20% fat content provided by: (1) Partially hydrogenated soybean oil (13% trans fats). (2) Soybean oil (44% PUFAs). (3) Palm oil (50% saturated fats). (4) Canola oil (49% MUFAs).	Plasma CETP activity	No effect
			Paraoxonase activity	No effect.
Vega-López S, USA (2009) [70]	30 postmenopausal women with moderate hypercholesterolemia Mean age ± SD: 64.2 ± 7.5 years	5-week crossover interventions with 20% calories from: (1) Corn oil. (2) Partially hydrogenated soybean oil.	Plasma LCAT activity	No effect.

SFA: saturated fatty acids. TFA: trans fatty acids. MUFA: monounsaturated fatty acids. PUFA: polyunsaturated fatty acids. DHA: docosahexaenoic acid. EPA: eicosapentaenoic acid. CEC: cholesterol efflux capacity. CETP: cholesteryl ester transfer protein. CETA: cholesteryl ester transfer activity. LCAT: lecithin–cholesterol acyltransferase. ABDP: apolipoprotein B-depleted plasma. ABDS: apolipoprotein B-depleted serum. PON1: paraoxonase-1. TBARS: thiobarbituric-acid-reactive substance. HOII: HDL oxidative/inflammatory index.

**Table 5 jcm-10-05897-t005:** Studies with dietary cholesterol.

First Author, Location (Year)	Study Participants	Intervention	HDL Function Analyzed	Results
Andersen CJ, USA (2013) [75]	37 metabolic syndrome patients (12 men and 25 women) Age range: 30–70 years	12-week parallel diet interventions with: (1) 3 whole eggs (534 mg cholesterol). (2) Equivalent egg substitute (without cholesterol).	^3^H CEC in RAW 264.7 cells in isolated HDLs	Increase of 2.4% in egg group relative to baseline.
Blanco-Molina A, Spain (1998) [73]	15 healthy men Mean age ± SD: 23.4 ± 5.6 years	24-day crossover diets with: (1) Low-fat NCEP Step I diet supplemented with two eggs. (2) Low-fat NCEP-Step I diet without eggs. (3) MUFA-rich diet supplemented with two eggs (4) MUFA-rich diet without eggs.	^3^H CEC in Fu5AH cells in serum	Increase in low-fat diet enriched with eggs compared to the low-fat diet without eggs.
Blesso CN, USA (2013) [82]	37 etabolic syndrome patients (12 men and 25 women) Mean age ± SD: 51.9 ± 7.7 years	12-week parallel carbohydrate-restricted diet interventions with: (1) Three whole eggs/day (534 mg cholesterol). (2) Yolk-free eggs.	Plasma CETP activity	No effect.
			Plasma LCAT activity	Increase in LCAT in whole egg group relative to baseline.
Ginsberg HN, USA (1995) [81]	13 healthy women Mean age ± SD: 23.5 ± 1.9 years	8-week crossover diets with: (1) One egg. (2) Two eggs. (3) Three eggs.	Plasma CETP mass	No effect.
Ginsberg HN, USA (1994) [80]	20 healthy men Mean age ± SD: 24.4 ± 2.7 years	8-week crossover low-fat diets with: (1) No eggs. (2) One egg. (3) Two eggs. (4) Four eggs.	Plasma CETP mass	4 eggs/day increased CETP levels by 6% compared to other diet interventions.
Herron KL, USA (2004) [78]	52 healthy participants (25 men and 27 women) Age range: 18–50 years	1-month crossover diets with: (1) Eggs (640 mg/day cholesterol). (2) Placebo egg substitute.	Plasma CETP activity	Increased CETP activity in egg group compared to control in a subgroup of hyper-responders to dietary cholesterol.
			Plasma LCAT activity	Increased LCAT activity in egg group compared to baseline in a subgroup of hyper-responders to dietary cholesterol.
Herron KL, USA (2003) [77]	40 normolipidemic men Age range: 20–50 years	1-month crossover diets with: (1) Eggs (640 mg/day cholesterol). (2) Placebo egg substitute.	Plasma CETP activity	Increased CETP activity in egg group compared to control in a subgroup of hyper-responders to dietary cholesterol.
			Plasma LCAT activity	Increased LCAT activity in egg group compared to control in a subgroup of hyper-responders to dietary cholesterol.
Herron KL, USA (2002) [76]	51 premenopausal women Age range: 19–49 years	1-month crossover diets with: (1) Eggs (640 mg/day cholesterol). (2) Placebo egg substitute.	Plasma CETP activity	Increased CETP activity in egg group compared to control in a subgroup of hyper-responders to dietary cholesterol.
Martin LJ, USA (1993) [79]	30 healthy men Mean age ± SD: 23.0 ± 2.6 years	35-day crossover intervention with: (1) Low-cholesterol diet (80 mg/1000 Kcal). (2) High-cholesterol diet (320 mg/1000 Kcal).	Plasma CETP mass	Increased levels in high-cholesterol diet compared to low cholesterol diet.
Missimer, USA (2018) [83]	50 healthy young participants (24 men and 26 women) Mean age ± SD: 23.3 ± 3.1 years	4-week crossover diets with: (1) Two large eggs/day (370 mg cholesterol). (2) Oatmeal (384 g/day).	Plasma CETP activity	No effect.
Morgantini, Italy (2018) [87]	14 healthy participants Mean age ± SD: 25.0 ± 2.3 years	2-week crossover intervention with: (1) Low-fat and low-cholesterol diet (100–150 mg/day; 5–10% SFA). (2) High-fat and high-cholesterol diet (250–300 mg/day; 15–20% SFA).	Paraoxonase activity	No effect.
			HDL hydroperoxides content	Increase in hydroperoxide content compared to low-fat and low-cholesterol diet.
			HDL associated SAA	Increase in SAA content in HDL compared to low-fat and low-cholesterol diet.
Mutungi G, USA, (2010) [86]	31 overweight or obese men Age range: 40–70 year	12-week parallel carbohydrate-restricted diets with: (1) Three liquid eggs. (2) Substitute egg placebo.	LCAT activity	Increase in egg group relative to control.
Sawrey-Kubicek, USA, (2019) [74]	20 overweight women Mean age ± SD: 57.7 ± 5.3 years	4-week crossover diet with: (1) Two whole eggs per day (100 g/egg). (2) Two yolk-free eggs per day (100 g/egg).	BODIPY-cholesterol-marked CEC in J774 cells in ABDP samples	Increase of 5.69% in whole egg group compared to control.
			Plasma CETP activity	No effect.
			Plasma LCAT activity	No effect.
			Plasma paraoxonase-1 activity	No effect.
Vorster HH, South Africa (1992) [85]	70 young healthy men Age range: 18–19 years	Parallel diet interventions with measurements at 1, 5, 7 months with: (1) 3 eggs/week. (2) 7 eggs/week. (3) 14 eggs/week.	Plasma LCAT activity	Increased LCAT activity in 14 eggs/week group relative to 3 eggs/week group after 1 month (but not after 5 or 7 months).
Waters D, USA (2007) [84]	22 postmenopausal women Age range: 50–77 years	4-week crossover diets with: (1) Eggs (640 mg/day cholesterol and 600 μg of lutein+zeaxanthin). (2) Placebo egg substitute.	Plasma CETP activity	No effect.

MUFA: monounsaturated fatty acids. CEC: cholesterol efflux capacity. CETP: cholesteryl ester transfer protein. LCAT: lecithin–cholesterol acyltransferase. ABDP: apolipoprotein B-depleted plasma. SAA: serum amyloid A.

**Table 6 jcm-10-05897-t006:** Studies with antioxidant nutrients and antioxidant-rich foods.

First Author, Location (Year)	Study Participants	Intervention	HDL Function Analyzed	Results
Balsan, Brazil (2019) [32]	142 overweight or obese participants (55 men and 87 women) Mean age ± SD: 50.2 ± 6.4 years	8-week parallel interventions with 1 L: (1) Mate tea. (2) Green tea. (3) Apple tea (control).	Serum PON1 mass	Higher levels of PON1 in Mate group compared to green tea and apple tea.
Baralic I, Serbia (2012) [98]	40 male soccer players Mean age ± SD: 17.91 ± 0.16 years	3-month parallel interventions with supplements of: (1) 4 mg of Astaxanthin. (2) Placebo.	Plasma PON1 paraoxonase and diazonase activity	Increase in diazonase activity relative to baseline levels.
Boaventura, Brazil (2012) [107]	74 dyslipidemic participants (17 men and 57 women) Mean age ± SD: 48.5 ± 11.6 years	3-month parallel interventions with 1 L: (1) Mate tea (1 L/day) (2) Low fat and vegetable rich diet. (3) Diet and mate tea.	Serum Arylesterase activity	No effect.
Bub A, Germany (2005) [100]	22 healthy young participants Mean age ± SD: 29 ± 6 years	2-week crossover interventions with 330 mL/day: (1) Tomato juice. (2) Carrot juice.	Serum Arylesterase activity	No effect.
Bub A, Germany (2002) [99]	50 elderly participants (18 men and 32 women) Mean age ± SD: 70 ± 6 years	8-week crossover interventions with 330 mL/day: (1) Tomato juice. (2) Water.	Serum Arylesterase activity	Both interventions increased relative to baseline levels.
Cherki M, Morocco (2005) [102]	60 healthy men Mean age ± SD: 23.4 ± 3.85 years	3-week parallel interventions with two oils rich in phenolic compounds: (1) Virgin argan oil. (2) Virgin olive oil.	Paraoxonase and arylesterase activity	Both interventions increased relative to baseline levels.
Dalgård C, Denmark (2007) [110]	48 participants with peripheral artery disease (35 men and 13 women) Mean age ± SD:61 ± 6 years	4-week parallel interventions with: (1) Vitamin E (15 mg/day) combined with orange and blackcurrant juice. (2) Placebo and combined with orange and blackcurrant juice. (3) Vitamin E combined with control juice (4) Placebo and control juice	Paraoxonase1 activity and mass	No effect.
De Roos B, The Netherlands (2000) [94]	46 healthy participants (23 men and 23 women) Mean age ± SD: 29.5 ± 2 years	24-week parallel interventions with 0.9 L/day: (1) French-press coffee. (2) Filtered coffee.	Serum CETP activity	French-press coffee increased CETP activity relative to filtered coffee.
Farràs M, Spain (2018) [89]	33 hypercholesterolemic participants (19 men and 14 women) Mean age ± SD: 55.2 ± 10.6 years	3-week crossover interventions with 25 mL of virgin olive oil per day: (1) Enriched with olive oil phenolic components (500 ppm) (FVOOT). (2) Enriched with olive oil phenolic components and other phenolic components from thyme (500 ppm in the aggregate) (FVOO). (3) Not enriched (VOO).	^3^H CEC in J774 cells in isolated HDL samples	Increase in CEC in FVOOT relative to FVOO.
			HDL antioxidant compounds	Increase in β-criptoxanthin and lutein in both enriched olive oils relative to baseline.
Farràs M, Spain (2015) [95]	33 hypercholesterolemic participants (19 men and 14 women) Mean age ± SD: 55.2 ± 10.6 years	3-week crossover interventions with 25 mL of virgin olive oil per day: (1) Enriched with olive oil phenolic components (500 ppm) (FVOOT). (2) Enriched with olive oil phenolic components and other phenolic components from thyme (500 ppm in the aggregate) (FVOO). (3) Not enriched (VOO).	Plasma CETP activity	No effect.
			Plasma LCAT mass	Increase in mass in FVOOT relative to VOO.
			Plasma PON1 arylesterase activity	Increase in FVOOT relative to VOO.
Fernández-Castillejo S, Spain (2017) [103]	33 hypercholesterolemic participants (19 men and 14 women) Mean age ± SD: 55.2 ± 10.6 years	3-week crossover interventions with 25 mL of virgin olive oil per day: (1) Enriched with olive oil phenolic components (500 ppm) (FVOOT). (2) Enriched with olive oil phenolic components and other phenolic components from thyme (500 ppm in the aggregate) (FVOO). (3) Not enriched (VOO).	Serum PON1 and PON3 mass and paraoxonase-1 and lactonase specific activity	FVOOT increase PON1 levels relative to baseline. FVOO increase paraoxonase-1 and lactonase activity relative to baseline levels. VOO increase PON3 mass relative to FVOO and FVOOT.
Freese R, Finland (2002) [96]	77 healthy participants Mean age (range age): 25.1 (19–52) years	6-week parallel dietary interventions with: (1) Low vegetable diet with high linoleic acid content. (2) High vegetable and apple diet with high linoleic content. (3) Low vegetable diet with high oleic acid content. (4) High vegetable and apple diet with high oleic acid content.	Plasma CETP activity	Increased CETP activity in high vegetables and linoleic group relative to baseline.
			Plasma LCAT activity	No effect.
Hernáez A, Spain (2014) [88]	47 healthy men Mean age ± SD: 33.5 ± 10.9 years	3-week crossover interventions with 25 mL raw olive oil per day containing: (1) Polyphenol-rich oil (366 mg/kg polyphenols). (2) Polyphenol-poor oil (2.7 mg/kg).	^3^H CEC in THP-1 cells in ABDS samples	Increase of 3.04 ± 9.98% relative to polyphenol-poor group.
			Polyphenol metabolites in HDL	Increased content of polyphenol metabolites in intervention group. compared to baseline.
Lazavi F, Iran (2018) [101]	42 diabetes type 2 participants (15 men and 27 women) Mean age ± SD: 56.86 ± 8.47 years	8-week parallel interventions with 200 mL/day: (1) Barberry Juice (2) Control.	Plasma PON1 concentration	Increase relative to control group.
McEneny J, UK (2013) [91]	54 moderate overweight participants Mean age ± SD: 50.4 ± 3.0 years	12-week parallel interventions with: (1) Lycopene-rich diet (224–350 mg/day). (2) Lycopene supplements (70 mg/day). (3) Control (placebo).	Serum CETP activity	Decrease in Lycopene supplement relative to lycopene diet and control.
			Serum LCAT activity	Increase in both lycopene interventions relative to baseline.
			PON1 arylesterase activity	Increase in both lycopene interventions relative to control. Increase in Lycopene supplement relative to lycopene diet.
			SAA mass in isolated HDL 2 and HDL3	Decrease in both lycopene interventions relative to control.
Michaličková, Czech Republic (2019) [108]	26 hypertensive participants (7 men and 19 women) Mean age: 47 years	4-week parallel interventions with 200 g tomato juice: (1) Lycopene and polyphenol rich. (2) Control juice.	Serum paraoxonase-1 activity	No effect.
Millar, USA (2018) [90]	20 Metabolic syndrome participants (12 men and 8 women) Mean age ± SD: 53.5 ± 10.1 years	3-week crossover interventions with: (1) 60 g/day freeze-dried grape powder. (2) Placebo.	PON1 arylesterase activity	No effect.
Ozdemir B, Turkey (2008) [105]	48 participants with hyperlipidemia (15 men and 33 women) Age range: 25–60 years	3-month parallel interventions: (1) Origanum onites aqueous distillate (75 mL/day). (2) Low-fat diet.	Paraoxonase and arylesterase activity	Increase relative to control.
Puglisi MJ, USA (2009) [93]	34 healthy participants (17 men and 17 women) Age range: 50–70 years	6-week parallel interventions: (1) Origanum onites aqueous distillate (75 mL/day) (2) Low-fat diet.	Plasma CETP activity	No effect.
Qian Q, China (2012) [106]	54 participants with type 2 diabetes and chronic heart disease Mean age ± SD: 59.8 ± 8.7 years	2-month parallel interventions: (1) Salvia hydrophilic extract (10 g/day) with diet and hypoglycemic drugs. (2) Diet and hypoglycemic drugs (control).	Paraoxonase activity	9% increase compared to baseline levels.
Qin Y, China (2009) [31]	120 participants with dyslipidemia (42 men and 78 women) Age range: 40–65 years	12-week parallel interventions with supplements: (1) Anthocyanins (320 mg/day). (2) Placebo.	^3^H CEC in J774 cells in serum	Anthocyanin group increased 20% relative to placebo.
			Plasma CETP activity and mass	Decreased mass and activity relative to placebo.
			Plasma LCAT activity and mass	No effect.
Shidfar F, Iran (2015) [104]	50 participants with type 2 diabetes Mean age ± SD: 45.2 ± 7.64 years	3-month parallel interventions with supplements: (1) 3 g of powdered ginger capsules daily. (2) Placebo.	Paraoxonase-1 activity	Increase relative to control.
Suomela JP, Finland (2006) [109]	14 healthy men Mean age ± SD: 47.2 ± 9.7 years	4-week crossover interventions with supplements: (1) 185 g sea-buckthorn-flavonol-enriched oatmeal porridge (78 mg flavonol). (2) Control porridge.	Paraoxonase-1 activity	No effect.
Van Tol A, The Netherlands (1997) [97]	10 healthy males Mean age ± SD: 24 ± 4 years	4-week crossover interventions with coffe-extract supplements: (1) 64 mg cafestol + 1 mg kahweol per day. (2) 60 mg cafestol + 54 mg kahweol per day.	Serum LCAT activity	Decrease of 11 ± 12% in cafestol + kahweol relative to baseline.
Zern TL, USA (2005) [92]	44 premenopausal or postmenopausal women Mean age ± SD: 39.7 ± 8.5 (premenopausal); 58.5 ± 7.5 (postmenopausal)	4-week crossover interventions with supplements: (1) 6 g/day grape powder. (2) Placebo.	Plasma CETP activity	Decrease relative to baseline levels (9% in premenopausal women and 29% in postmenopausal).
Zhu Y, China (2013) [25]	122 hypercholesterolemic participants (50 men and 72 women) Age range: 40–65 years	24-week parallel interventions with supplements: (1) Anthocyanins (320 mg/day). (2) Placebo.	^3^H CEC in J774 cells in isolated HDL	Anthocyanin group increased 17.7% relative to placebo.
			Paraoxonase-1 activity	Anthocyanin group increased 17.4% relative to placebo.

CEC: cholesterol efflux capacity. CETP: cholesteryl ester transfer protein. LCAT: lecithin–cholesterol acyltransferase. ABDP: apolipoprotein B-depleted plasma. SAA: serum amyloid A. PON1: paraoxonase-1. PON3: paraoxonase-3.

**Table 7 jcm-10-05897-t007:** Studies with antioxidant-rich dietary patterns.

First Author, Location (Year)	Study Participants	Intervention	HDL Function Analyzed	Results
Damasceno NR, Spain (2013) [28]	169 participants at high cardiovascular risk (74 men and 95 women) Mean age: 67 years	1-year parallel whole-diet interventions: (1) Traditional Mediterranean diet enriched with extra virgin olive oil (1 L/week). (2) Traditional Mediterranean diet enriched with nuts (30 g/day of mixed nuts). (3) Low-fat diets.	Serum CETP activity	A traditional Mediterranean diet enriched with olive oil and a low-fat diet decreased CETP levels compared to baseline.
Daniels JA, UK (2014) [111]	74 obese participants with type 2 diabetes (52 men and 22 women) Age range: 40–70 years	8-week parallel whole-diet interventions: (1) Low fruit and vegetable intake (80 g/day). (2) High fruit and vegetable intake (400 g/day).	Serum LCAT activity	Increase in fruit-rich and vegetable-rich diets relative to baseline
			PON1 arylesterase activity in Serum and in HDL2 and HDL3	Increase in serum activity in vegetable-rich and fruit-rich diets relative to baseline. Increase in HDL3 in vegetable-rich group relative to low-vegetable group.
			SAA content in HDL2 and HDL3	No effect
			HDL2 and HDL3 content in carotenoids	Vegetable-rich group increased HDL3 α-carotene, β-cryptoxanthin, lutein, and lycopene compared to low vegetable intake. HDL2 intervention increased β-cryptoxanthin compared to control, and lutein relative to baseline.
Hernáez Á, Spain (2020) [21]	358 participants at high cardiovascular risk (131 men and 227 women) Mean age: 67 years	1-year parallel whole-diet interventions: (1) Traditional Mediterranean diet enriched with extra virgin olive oil (1 L/week). (2) Traditional Mediterranean diet enriched with nuts (30 g/day of mixed nuts). (3) Low-fat diets.	ABDP HDL-alpha-1-antitrypsin	Decrease in Mediterranean diet with olive oil compared to baseline.
			Nitric oxide production in HUVEC cells after ABDP.	Increase in Mediterranean diet with virgin olive oil compared to low-fat diet.
Hernáez Á, Spain (2017) [19]	296 participants at high cardiovascular risk (151 men and 145 women) Mean age ± SD: 65.9 ± 6.43 years	1-year parallel whole-diet interventions: (1) Traditional Mediterranean diet enriched with extra virgin olive oil (1 L/week). (2) Traditional Mediterranean diet enriched with nuts (30 g/day of mixed nuts). (3) Low-fat diet.	^3^H CEC in THP-1 cells in ABDP	Both Mediterranean diets increased CEC relative to baseline levels.
			Plasma CETP activity	Mediterranean diet with virgin olive oil decreased CETP activity relative to baseline
			Direct HDL antioxidant capacity on LDL	Increased antioxidant capacity after a Mediterranean diet with olive oil relative to baseline.
			HDL oxidation status by TBARS assay	Decreased oxidation status relative to baseline levels in Mediterranean diet with olive oil and in low-fat diet.
			HDL oxidative/inflammatory index (HOII)	The control low-fat diet increased HOII relative to baseline levels.
			Serum PON1 arylesterase activity	Mediterranean diet with virgin olive oil increased PON1 activity relative to low-fat diet
			Nitric oxide production in HUVEC cells after ABDP.	Increase in Mediterranean diet with virgin olive oil compared to low-fat diet.
Rantala M, Finland (2002) [112]	37 healthy women Mean age ± SD: 42.6 ± 10.1 years	5-week parallel whole-diet interventions: (1) Low-vegetable diet (1 serving/day) (2) Vegetable-rich diet (430 mg of vitamin C, 18 mg of carotenoids, 17 mg of vitamin E and 600 g of folate.)	Paraoxonase-1 activity	Vegetable-rich diet decreased PON activity compared to low-vegetable diet.

CEC: cholesterol efflux capacity. CETP: cholesteryl ester transfer protein. LCAT: lecithin–cholesterol acyltransferase. ABDP: apolipoprotein B-depleted plasma. SAA: serum amyloid A. PON1: paraoxonase-1. HOII: HDL oxidative/inflammatory index.

**Table 8 jcm-10-05897-t008:** Studies on ethanol and HDL function.

First Author, Location (Year)	Study Participants	Intervention	HDL Function Analyzed	Results
Beulens JW, The Netherlands (2004) [116]	24 healthy men Mean age ± SD: 52 ± 5 years	17-day crossover interventions with: (1) 40 g/day ethanol (whisky). (2) Water (control).	^3^H CEC in J774 and Fu5AH cells in serum	CEC increased in both cellular models relative to control water group.
Králová Lesná I, Cezch Rep. (2010) [118]	13 healthy men Mean age ± SD: 32.31 ± 5.9 years	4-week crossover interventions with: (1) 36 g alcohol/day (1L beer). (2) Control abstinence period.	^14^C CEC in THP-1 cells in plasma	No effect.
Padro T, Spain (2018) [117]	36 overweight or obese I, regular moderate alcohol consumers (21 men and 15 women) Mean age ± SD: 48.3 ± 5.4 years	4-week crossover interventions with: (1) Beer (men, 2 cans; women, 1 can; 15 g/can ethanol and 604 mg/can polyphenols). (2) Non-alcoholic beer (414 mg polyphenols/can).	^3^H CEC in J774 cells in ABDS	Increase in alcoholic beer group relative to baseline levels.
			HDL Antioxidant Potential assessed by TRAP test	Both groups increased antioxidant capacity of HDLs relative to baseline.
Senault C, France (2000) [113]	56 healthy young men	2-week parallel interventions with: (1) Red wine (30 g alcohol/day). (2) A solution with the same degree of alcohol as red wine (30 g alcohol/day). (3) Control alcohol-free red wine.	^3^H CEC in Fu5AH cells in serum	Increase of 7% relative to baseline.
			Plasma CETP activity	No effect.
Sierksma A, The Netherlands (2004) [115]	18 healthy women Mean age ± SD: 57 ± 5 years	3-week crossover interventions with: (1) 24 g/day ethanol (white wine). (2) Grape juice control.	^3^H CEC in Fu5AH cells in plasma	Increase of 3.4% relative to control.
			Plasma Cholesteryl ester transferactivity	No effect.
Sierksma A, The Netherlands (2002) [120]	19 healthy participants (10 men and 9 women). Age range:45–64 years	3-week crossover interventions with: (1) 30–40 g/day ethanol (beer). (2) No alcohol control.	Serum PON paraoxonase-1 activity and PON mass	Increase in PON-1 activity and mass in beer group relative to control.
Van der Gaag MS, The Netherlands (2001) [114]	11 healthy men Age range: 45–60 years	3-week crossover interventions with 40 g/day ethanol: (1) Red wine. (2) Beer. (3) Spirits (Dutch gin). (4) Water control.	^3^H CEC in Fu5AH cells in plasma	Red wine increased CEC by 5%, beer by 6.9% and gin by 6%, all relative to control.
Van der Gaag MS, The Netherlands (1999) [119]	11 healthy men Age range: 45–60 years	3-week crossover interventions with 40 g/day ethanol: (1) Red wine. (2) Beer. (3) Spirits (Dutch gin). (4) Water control.	Paraoxonase-1 activity	Red wine increased PON-1 activity by 6.9%, beer by 7.4% and gin by 9.3%, all relative to control.

CEC: cholesterol efflux capacity. CETP: cholesteryl ester transfer protein. LCAT: lecithin–cholesterol acyltransferase. ABDS: apolipoprotein B-depleted serum. PON1: paraoxonase-1. HOII: HDL oxidative/inflammatory index. TRAP: total radical-trapping antioxidative potential.

**Table 9 jcm-10-05897-t009:** Studies with physical activity, calorie restriction, and HDL function.

First Author, Location (Year)	Study Participants	Intervention	HDL Function Analyzed	Results
Albaghdadi MS, USA (2017) [18]	88 participants with peripheral artery disease (41 men and 47 women) Mean age ± SD: 70.85 ± 1.72 years	24-week parallel interventions with: (1) Endurance (treadmill) (3 times/w). (2) Strength (lower-extremity resistance-training group) (3 times/w). (3) Control-diet group.	^3^H CEC in J774 cells in ABDS	No effect.
Dokras A, USA (2018) [124]	87 overweight or obese women with polycystic ovary syndrome Age range: 18–40 years	16-week parallel interventions with: (1) Oral contraceptive pills. (2) Recommendations for calorie restriction of 500-calorie deficit + brisk walking 5 times/week). (3) Combined treatment.	^3^H CEC in J774 cells in ABDS	No effect.
Khan AA, Australia (2018) [123]	53 metabolic syndrome patients (30 men and 23 women) Mean age ± SD: 55 ± 6 years	12-week parallel interventions with: (1) Aerobic-training excercise 4 times per week for 30–40 min combined with diet (DASH) (to reduce by 600 kcal/day). (2) DASH dietary intervention. (3) Control group without intervention.	^3^H CEC in THP-1 cells in ABDS	DASH diet combined with exercise increased CEC by 25% relative to baseline.
			Plasma CETP activity	DASH diet combined with exercise decreased CETP relative to baseline.
Miida T, Japan (1998) [126]	24 hypercholesterolemic and 12 normolipidemic participants (9 men and 27 women) Mean age ± SD: 57.93 ± 8.37 years	4-week parallel interventions with: (1) Low-calorie diet NCEP Step I. (2) Probucol 500 mg/day. (3) Probucol 1000 mg/day. (4) Control with normolipidemic patients.	ABDP CETP mass	No effect.
Rönnemaa T, Finland (1988) [128]	25 diabetic participants Mean age: 52.7 years	4-month parallel interventions with: (1) Aerobic exercise 5–7 days/week, 45 min/session, 70% of VO2 Max.). (2) No-training group.	Serum LCAT activity	No effect
Sarzynski, USA (2018) [20]	Participants from STRRIDE-PD trial: 106 overweight sedentary (23 men and 67 women) Age range:18–65 years	6-month parallel interventions with endurance training: (1) Low levels of moderate-intensity exercise. (2) High levels of moderate-intensity exercise. (3) High levels of vigorous exercise. (4) Low-fat diet combined with moderate-intensity exercise.	^3^H and BODIPY CEC in J774 cells in ABDP	Increase in ^3^H CEC in high levels of exercise compared to the other three interventions. No effect with BODIPY-marked CEC.
	Participants from E-MECHANIC trial: 90 overweight sedentary (39 men and 67 women) Age range: 45–75 years	6-month parallel weight-loss interventions with: (1) Low levels of moderate-intensity exercise (to reduce 8 kcal/kg weeks). (2) High levels of exercise (to reduce 20 kcal/kg weeks). (3) No-exercise group (control).	^3^H and BODIPY CEC in J774 cells in ABDP	Increase in ^3^H non-ABCA1 CEC in high levels exercise compared to control group. No effect with BODIPY-marked CEC.
Talbot, The Netherlands (2018) [122]	77 overweight/obese participants Age range: 18–65 years	6-week parallel interventions with: (1) Very-low-calorie diet (500 kcal). (2) Control group without weight loss.	BODIPY CEC in J774 cells in ABDP	No effect.
			Cholesterol ester transfer from radio-labeled HDL to ApoB lipoproteins.	No effect.
Thomas TR, USA (1985) [127]	36 young healthy men Age range:18–25 years	11-week parallel interventions with: (1) 3 times/week 5 miles continuous exercise with 4-minutes interval (1:1, work:rest). (2) 3 times/week 5 miles continuous exercise with 2-minute intervals (1:1-1/2, work:rest). (3) No-training group.	LCAT levels	No effect.
Tiainen S, Finland (2016) [22]	161 sedentary women Age range: 43–63 year	6-month parallel interventions with: (1) Aerobic training four times/week. (2) Control without exercise.	CETP activity	No effect.
Vislocky LM, USA (2007) [125]	12 healthy unfit participants (7 men and 5 women) Age range:18–30 years	8-week parallel interventions with: (1) 12 eggs/week. (2) No eggs. (3) Endurance training 30–45 min. 3–5 days/week. (4) No-training group (control).	Plasma CETP activity	32% decrease in trained participants relative to untrained participants.
Wesnigk J, Belgium (2016) [23]	16 obese adolescents Mean age ± SD: 15.1 ± 2.5 years	10-month parallel interventions with: (1) Dietary restriction of 1500–1800 kcal/day combined with intensive supervised exercise (2 h/day of lifestyle activities + 3 times/week 40’ aerobic and resistance training) and psychological support from experts. (2) Usual-care group (control).	^3^H CEC in J774 cells in ABDS	Increase relative to usual-care group.
			eNOS phosphorilation mediated by HDL in HAECs cells	Increase relative to usual-care group.
Williams PT, USA (1990) [27]	77 healthy sedentary men Age range:30–55 years	1-year parallel interventions with: (1) Running group (12.7 km/week on treadmill). (2) No-training group.	Plasma LCAT mass	No effect.
Woudberg NJ, South Afica (2018) [121]	35 obese black women Mean age ± SD: 24.5 ± 0.9 years	12-week parallel interventions with: (1) Exercise (combination of aerobic and resistance exercise 40–60 min., 4 days per week). (2) No-exercise group (control).	^3^H CEC in RAW264.7 cells in isolated HDL	No effect.
			serum PON1 activity	Decrease relative to control group.
			HDL-bound phospholipase A2 expression in HDLs	No effect.
			VCAM expression in isolated HDL	No effect.

CEC: cholesterol efflux capacity. CETP: cholesteryl ester transfer protein. LCAT: lecithin–cholesterol acyltransferase. ABDP: apolipoprotein B-depleted plasma. ABDS: apolipoprotein B-depleted serum. SAA: serum amyloid A. PON1: paraoxonase-1. TBARS: thiobarbituric-acid-reactive substance. eNOS: endothelial nitric oxide synthase. PAF-AH: platelet-activating factor acetylhydrolase. VCAM: vascular cell-adhesion protein.

**Table 10 jcm-10-05897-t010:** Other lifestyle interventions.

First Author, Location (Year)	Study Participants	Intervention	HDL Function Analyzed	Results
Favari E, Italy (2020) [129]	41 overweight participants Age range: 30–65 years	12-week parallel interventions with whole-wheat pasta enriched with phenolic acids (50.3 mg/100 g) + fiber (12.5 g/100 g): (1) Enriched with β-glucans (2.3 g/100 g) and Bacillus coagulans. (2) Non-enriched pasta.	^3^H CEC in CHO cells in serum	Increase in enriched group relative to control.
Higashi K, Japan (2001) [132]	14 healthy men Mean age ± SD: 31 ± 4 years	4-week crossover interventions with supplements: (1) 20 g per day of soy protein. (2) Placebo.	CETP mass	No effect.
			LCAT activity	No effect.
Homma Y, Japan (2003) [30]	105 healthy participants (38 men and 67 women) Mean age ± SD: 47 ± 13 years	4-week parallel interventions with: (1) 2 g/day plant stanol. (2) 3 g/day plant stanol. (3) Placebo.	Plasma CETP mass	Decrease of 6.1% after 2 g/day of stanol and 3.3% in 3 g/day relative to baseline.
Lichtenstein AH, USA (2002) [135]	36 participants with high levels of LDL-C (18 men and 18 women) Age range: 55–74 years	4.5-week crossover interventions with supplements: (1) TLC/Step 2 diet (low in saturated fats and rich in PUFA and fibre). (2) Western diet (high-fat diet).	ABDP CETP activity	No effect.
Lottenberg AM, Brazil (2003) [138]	60 moderately hypercholesterolemic participants (10 men and 50 women) Age range: 20–60 years	4-week crossover interventions with margarine (20 g/day): (1) Enriched with plant sterol ester (2.8 g/day equal to 1.68 g/day phytosterols). (2) Non-enriched (control).	Plasma CETP mass	Decrease relative to placebo.
			Plasma LCAT activity	No effect.
Meng, USA (2018) [130]	11 healthy participants (7 men and 4 women) Mean age ± SD: 65 ± 8 years	4.5-week crossover interventions with foods containing: (1) Simple carbohydrates. (2) Refined carbohydrates. (3) Unrefined carbohydrates.	^3^H CEC in PBMCs cells in isolated HDLs	No effect.
Richter CK, USA (2018) [135]	20 moderate hypertension participants (9 men and 20 women) Mean age ± SD: 51.6 ± 6.6 years	6-week crossover intervention with soya protein: (1) 50 g/day. (2) 25 g/day. (3) No-soya group (control).	^3^H CEC in J774 cells in ABDP.	No effect.
Shidfar F, Iran (2009) [133]	52 hypercholesterolemic postmenopausal women Age range: 49–61 years	10-week parallel interventions with: (1) Soy protein (50 g/day (164 mg isoflavones)). (2) Placebo.	Plasma Paraoxonase-1 activity	Increase in fiber group relative to control.
Shresth S, USA (2007) [136]	33 healthy participants (11 men and 22 women) Age range: 35–65 years	1-month crossover interventions with supplements: (1) 10 g Psyllium yielding 7.68 g/day soluble fiber and 2.6 g/day plant sterols. (2) Placebo.	Plasma CETP activity	Supplemented group presented 11% lower CETP. activity relative to placebo.
Vega-López S, USA (2001) [137]	68 healthy participants (24 men and 23 premenopausal and 21 postmenopausal women) Mean age ± SD: 43.7 ± 13.2 years	1-month crossover interventions with supplements: (1) 15 g/day of psyllium fiber. (2) Placebo.	Plasma CETP activity	Decrease relative to placebo intervention.
Wood RJ, USA (2006) [134]	30 overweight men Age range: 20–69 years	12-week parallel interventions with carbohydrate-restriction diets: (1) Supplemented with fiber (3 g/day of konjac-mannan fiber). (2) Non-supplemented control.	Plasma CETP activity	No effect.
			Plasma LCAT activity	Increase in fiber group relative to baseline.

CEC: cholesterol efflux capacity. CETP: cholesteryl ester transfer protein. LCAT: lecithin–cholesterol acyltransferase. ABDP: apolipoprotein B-depleted plasma.

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
