# Peer review of "Modification of High-Density Lipoprotein Functions by Diet and Other Lifestyle Changes: A Systematic Review of Randomized Controlled Trials"

_jcm, 2021, doi:10.3390/jcm10245897_

Round 1

Reviewer 1 Report

Authors performed a broad meta-analysis of dietary effects and physical activity effect on HDL functions. The manuscript is well planned and written, though I have some minor suggestions.

Firstly, I would strongly encourage to say something more about HDL and their role in lipoproteins/lipids metabolism. I miss this information and it is crucial for a reader to have an understanding where these HDL are coming from, how they are generated and what their role is in lipids metabolism. I think a great additive to the review would be a figure showing this metabolism, something like described recently in Cells https://doi.org/10.3390/cells10030597

Secondly, you mention 12 HDL functions and analyze how they were/were not modified by diet and/or physical activity but you do not explain anywhere the mechanism of these functions e.g., what is this CEC and what is the role of CETP, LCAT etc. I think that every paragraph you start describing dietary impacts/physical activity on a specific HDL function, this should be firstly explained - what is the mechanism of the described function and then results of your meta-analysis. It can be one or two sentences but please add this brief information.

Finally, please rewrite a your last two sentences in the first paragraph onf your conclusions so it is more clear what do you mean by ethanol enhancing "HDL functionality" and eggs increasing "cholesterol fractions". What exact function of HDL is increased by ethanol and what cholesterol fraction you mean, is it HDL-cholesterol, total cholesterol or what?

Author Response

REVIEWER 1 (reviewer’s comments in bold)

Authors performed a broad meta-analysis of dietary effects and physical activity effect on HDL functions. The manuscript is well planned and written, though I have some minor suggestions.

First, we would like to thank the reviewer for his/her comments made to our article. 

Firstly, I would strongly encourage to say something more about HDL and their role in lipoproteins/lipids metabolism. I miss this information and it is crucial for a reader to have an understanding where these HDL are coming from, how they are generated and what their role is in lipids metabolism. I think a great additive to the review would be a figure showing this metabolism, something like described recently in Cells https://doi.org/10.3390/cells10030597

Following the reviewer's comment, we have added a new Figure to provide an overview of the lipoproteins and lipid metabolism.

Figure 1. Lipoprotein metabolism overview. Lipid distribution in our body occurs in three different pathways. First, the exogenous pathway (red arrows): The absorption of dietary triglycerides, free cholesterol and cholesteryl esters are produced in small intestine. In enterocytes, dietary lipids are packed in chylomicrons and diffused to bloodstream. Chylomicrons diffuse triglycerides to peripheral cells and their remnants are, therefore, cleared in the liver. Second, the endogenous pathway (green arrows): Triglycerides and cholesterol synthetized in the liver are recirculated in bloodstream packed in VLDL. VLDL transports triglycerides to peripheral cells. VLDL remnants are transported to the liver, where the rest of triglycerides are removed by hepatic lipase action and become LDL. LDL transports cholesterol to peripheral tissues and are eventually cleared by the liver. Finally, HDLs are responsible for the reverse cholesterol transport (blue arrows): ApoA1 is synthetized in the liver and in enterocytes and released as a lipid free-monomer. Then it incorporates phospholipids by action of PLTP from VLDL. Lipid free ApoA1 is able to collect free cholesterol of peripheral cells, such macrophages, through ABCA1 receptor. The accumulation of phospholipids and free cholesterol results in the formation of discoidal HDL. Free cholesterol is transformed to cholesteryl esters and continuously internalized in HDL core by LCAT enzyme forming the mature form of HDL. Mature HDL continuous to pick up cholesterol through ABCG1 and SR-BI receptors. Finally, the cholesterol accumulated can be transported back to the liver mainly in an indirect way (exchanging cholesteryl esters for triglycerides with VLDL through CETP activity) or, in a minor proportion, directly through hepatic receptors. Figure was produced using Servier Medical Art (http://smart.servier.com/). ABCA1: ATP-binding cassette transporter A1. ABCG1: ATP-binding cassette transporter G1. ApoA1: Apolipoprotein A1 CE: cholesterol esters. CETP: Cholesteryl ester transfer protein. FC: free cholesterol. HDL: High density lipoprotein LCAT: Lecithin cholesterol acyltransferase. LDL: Low density lipoprotein. PL: Phospholipid. PLTP: Phospholipid transfer protein.TG: Triglycerides. VLDL: Very low density lipoprotein.

Secondly, you mention 12 HDL functions and analyze how they were/were not modified by diet and/or physical activity but you do not explain anywhere the mechanism of these functions e.g., what is this CEC and what is the role of CETP, LCAT etc. I think that every paragraph you start describing dietary impacts/physical activity on a specific HDL function, this should be firstly explained - what is the mechanism of the described function and then results of your meta-analysis. It can be one or two sentences but please add this brief information.

We agreed with the reviewer. The first draft of the review we provided a very little explanation of the mechanism involved in the 12 HDL functional traits. Nevertheless, we propose now to explain all the HDL functions in the introduction section to provide mechanistic data to better understand the metanalysis results.

Introduction section (paragraph 2, lines 77-112)

“HDL functions and associated functional components have been shown to be independently associated with lower CVD incidence [7] and stand as promising biomarkers to explain the HDL athero-protective role. The most studied HDL athero-protective function is the reverse cholesterol transport (Figure 1). It can be measured in vitro by the cholesterol efflux capacity (CEC) technique. It evaluates the ability of HDLs to remove cholesterol excess from cells and is measured in macrophage-derived cell cultures incubated with radio-labelled or fluorescent-labelled cholesterol [8]. HDLs can be also linked  with enzymes related HDL reverse cholesterol transport such lecithin cholesterol acyltransferase (LCAT), involved in cholesterol esterification, or cholesteryl ester transfer protein (CETP), crucial for cholesterol removal to the liver (see Figure 1) [8]. The second HDL athero-protective function is the antioxidant capacity, the ability to prevent LDL oxidation [9]. They carry antioxidant enzymes capable of degrading oxidized lipids, mainly paraoxonase-1 (PON1) and phospholipase A2 (LpPLA2) [9]. The global antioxidant capacity of HDL can be measured in vitro, by techniques such HDL oxidative/inflammatory index (HOII) [10]. On contrary, HDL could become dysfunctional after the oxidation of their components. The oxidative status of HDL can be evaluated by measuring its content of malondialdehyde (a lipid peroxide) [11]. A third HDL function is the capacity to modulate inflammatory responses. HDL is potentially able to decrease expression of endothelial adhesion proteins and chemokines [12]. However, HDL can also carry on their surface pro-inflammatory proteins related with HDL dysfunctionality, such serum amyloid A (SAA) and alpha-1-antitrypsin [13]. And finally, this lipoprotein could also present a protective effect on the endothelial layer of the arteries [12]. A healthy endothelium maintains a proper permeability and regulates vascular tone, helping to prevent atherosclerosis. In this regard, HDL has shown the capacity to improve the release of nitric oxide, a potent vasodilator secreted by the endothelium. The nitric oxide production can be evaluated in vitro in cellular cultures of endothelial cells [12].”

Finally, please rewrite a your last two sentences in the first paragraph onf your conclusions so it is more clear what do you mean by ethanol enhancing "HDL functionality" and eggs increasing "cholesterol fractions". What exact function of HDL is increased by ethanol and what cholesterol fraction you mean, is it HDL-cholesterol, total cholesterol or what?

We have rewritten the last two sentences in the first paragraph of the conclusion section.

Conclusion section (lines 1020-1026)

“In addition, reverse cholesterol transport with ethanol at moderate quantities, in studies mainly performed in healthy individuals, was able to enhance CEC. Finally, cholesterol dietary interventions with eggs increase circulant cholesterol fractions (total and HDL cholesterol) and, in concordance, increase CEC, CETP activity and LCAT activity.”

Reviewer 2 Report

This very extensive review contains a comprehensive overview of the topic. The authors should discuss certain limitations before the final conclusion based on the following considerations:    

1. Lines 85-94. I am not convinced that the different assays listed reflect HDL function. With regard to LCAT, I suppose the authors mean alpha-LCAT activity since beta-LCAT activity occurs in VLDL and LDL. Moreover, is CETP activity a function of HDL? It is indeed mainly bound to HDL but measurement of CETP activity is not corresponding to a biological function of HDL (anti-inflammation, endothelium protection, anti-oxidation, anti-apoptosis, anti-thrombosis, …) Why was phospholipid transfer protein (PLTP) activity not considered? These different enzymes modulate HDL metabolism and  HDL composition but do not directly reflect HDL function.

2. HDL function assays may have incremental value compared to classical biochemical tests. However, there are problems of standardization and measurement precision. It remains debatable for most of these assays whether they constitute a (valid) surrogate endpoint and adequately predict clinically relevant endpoints.  3. Does HDL function capture the effects of the interventions in different trials? 4. The HDL hypothesis is essentially unproven. For a detailed discussion, see  • PMID: 24236451   • DOI: 10.2174/1566524013666131118113927

Author Response

REVIEWER 2(reviewer’s comments in bold)

This very extensive review contains a comprehensive overview of the topic. The authors should discuss certain limitations before the final conclusion based on the following considerations:    

We would like to thank the reviewer for all the comments made to our article.

  1. Lines 85-94. I am not convinced that the different assays listed reflect HDL function. With regard to LCAT, I suppose the authors mean alpha-LCAT activity since beta-LCAT activity occurs in VLDL and LDL. Moreover, is CETP activity a function of HDL? It is indeed mainly bound to HDL but measurement of CETP activity is not corresponding to a biological function of HDL (anti-inflammation, endothelium protection, anti-oxidation, anti-apoptosis, anti-thrombosis, …) Why was phospholipid transfer protein (PLTP) activity not considered? These different enzymes modulate HDL metabolism and  HDL composition but do not directly reflect HDL function.

Firstly, we agreed that not all measurements are strictly HDL functions. The term “HDL functions” could be vague in some situations. The content of components such CETP, LCAT or SAA are functional components which can modify different HDL functions (either improving or worsening these functions).

Regarding your point, and to clarify this issue, we now explain all HDL functional traits in the abstract section, and changed the term “HDL function” by “HDL functions and associated functional components” or in the introduction and in the methods section where all the HDL traits are defined.

Abstract (lines 40-46)

“Results from HDL functions  and associated functional components including cholesterol efflux capacity, cholesteryl ester transfer protein, lecithin-cholesterol acyltransferase, HDL antioxidant capacity, HDL oxidation status, paraoxonase-1 activity, HDL anti-inflammatory and endothelial protection capacity, HDL-associated phospholipase A2, HDL-associated serum amyloid A, and HDL-alpha-1-antitrypsin were extracted.”

Introduction section (paragraph 2, lines 77-112)

“HDL functions and associated functional components have been shown to be independently associated with lower CVD incidence [7] and stand as promising biomarkers to explain the HDL athero-protective role. The most studied HDL athero-protective function is the reverse cholesterol transport (Figure 1). It can be measured in vitro by the cholesterol efflux capacity (CEC) technique. It evaluates the ability of HDLs to remove cholesterol excess from cells and is measured in macrophage-derived cell cultures incubated with radio-labelled or fluorescent-labelled cholesterol [8]. HDLs can be also linked  with enzymes related HDL reverse cholesterol transport such lecithin cholesterol acyltransferase (LCAT), involved in cholesterol esterification, or cholesteryl ester transfer protein (CETP), crucial for cholesterol removal to the liver (see Figure 1) [8]. The second HDL athero-protective function is the antioxidant capacity, the ability to prevent LDL oxidation [9]. They carry antioxidant enzymes capable of degrading oxidized lipids, mainly paraoxonase-1 (PON1) and phospholipase A2 (LpPLA2) [9]. The global antioxidant capacity of HDL can be measured in vitro, by techniques such HDL oxidative/inflammatory index (HOII) [10]. On contrary, HDL could become dysfunctional after the oxidation of their components. The oxidative status of HDL can be evaluated by measuring its content of malondialdehyde (a lipid peroxide) [11]. A third HDL function is the capacity to modulate inflammatory responses. HDL is potentially able to decrease expression of endothelial adhesion proteins and chemokines [12]. However, HDL can also carry on their surface pro-inflammatory proteins related with HDL dysfunctionality, such serum amyloid A (SAA) and alpha-1-antitrypsin [13]. And finally, this lipoprotein could also present a protective effect on the endothelial layer of the arteries [12]. A healthy endothelium maintains a proper permeability and regulates vascular tone, helping to prevent atherosclerosis. In this regard, HDL has shown the capacity to improve the release of nitric oxide, a potent vasodilator secreted by the endothelium. The nitric oxide production can be evaluated in vitro in cellular cultures of endothelial cells [12].”

Methods, section 2.1 (lines 171-179)

“We performed twelve searches, one for each HDL function or HDL associated functional components terms; 1) CEC activity; 2) CETP activity; 3) lecithin cholesterol acyl transferase (LCAT) activity; 4) HDL antioxidant capacity; 5) HDL oxidation status; 6) PON1 activity; 7) HDL anti-inflammatory and endothelial protection properties; 8) HDL-associated phospholipase A2; 9) HDL-associated with SAA; 10) HDL sphingosine-1-phosphate content; 11) HDL-alpha-1-antitrypsin; and 12) HDL-associated complement proteins.”

With regard to LCAT point, unfortunately the articles did not specify if the activity is alpha or beta. We have added this fact as a study limitation.

Discussion section (lines 1001-1003)

“Fourth, most studies did not specify whether the LCAT activity is beta or alpha, and consequently, if the activity is specifically linked to HDL.”

Finally, we would like to clarify that we also considered the fact of adding PLTP in systematic review, but there was a lack of lifestyle interventions evaluating PLTP activity and, as the review was already very extensive, we focused on the most studied HDL functions and characteristics.

  1. HDL function assays may have incremental value compared to classical biochemical tests. However, there are problems of standardization and measurement precision. It remains debatable for most of these assays whether they constitute a (valid) surrogate endpoint and adequately predict clinically relevant endpoints. 

Following the reviewer comment, we have added a limitation in discussion section (lines 992-999):

“Second, there is a marked heterogeneity in the laboratory procedures to evaluate HDL functions. For example, to evaluate CEC the analyzed studies employed 6 different types of cell cultures (J774, THP-1, Fu5AH, CHO, RAW264.7, and PBMCs cells) and three different types of labeled cholesterol (2 radio-labeled and 1 fluorescent). Some of these assays could hinder problems of standardization and measurement precision and remains to be proved whether they constitute adequate surrogate endpoints.”

  1. Does HDL function capture the effects of the interventions in different trials?

The term HDL function is extensive, as we have explained now in the introduction, we have selected 12 biomarkers related to the HDL functionality based on the existence of scientific papers and we consider that these markers constitute a broad batery of analysis to describe both terms, HDL properties and composition related to HDL functionality in human studies.

  1. The HDL hypothesis is essentially unproven. For a detailed discussion, see  • PMID: 24236451   • DOI: 10.2174/1566524013666131118113927

We agreed that HDL function role on cardiovascular diseases needs to be proven. Nevertheless, the scope of our review is to evaluate the effect of lifestyle interventions on HDL functions and functional components and provides some possible mechanism involved. 

To better clarify this issue we have soften some statements.

Abstract (lines 32-33) (using may instead of can)

High-density lipoprotein (HDL) functional traits have emerged as relevant elements that may explain HDL anti-atherogenic capacity better than HDL cholesterol levels.”

Introduction (lines 77-79)

“On the other hand, HDL functions and associated functional components have been shown to be independently associated with lower CVD incidence [7] and stand as promising alternative biomarkers to explain the HDL athero-protective role.”